# Repeated evolution of self-compatibility for reproductive assurance

Bart P.S. Nieuwenhuis[1,2], Sergio Tusso[1,2], Pernilla Bjerling[3], Josefine Stångberg[1], Jochen B.W. Wolf[1,2] & Simone Immler[1,4]

Sexual reproduction in eukaryotes requires the fusion of two compatible gametes of opposite sexes or mating types. To meet the challenge of finding a mating partner with compatible gametes, evolutionary mechanisms such as hermaphroditism and self-fertilization have repeatedly evolved. Here, by combining the insights from comparative genomics, computer simulations and experimental evolution in fission yeast, we shed light on the conditions promoting separate mating types or self-compatibility by mating-type switching. Analogous to multiple independent transitions between switchers and non-switchers in natural populations mediated by structural genomic changes, novel switching genotypes readily evolved under selection in the experimental populations. Detailed fitness measurements accompanied by computer simulations show the benefits and costs of switching during sexual and asexual reproduction, governing the occurrence of both strategies in nature. Our findings illuminate the trade-off between the benefits of reproductive assurance and its fitness costs under benign conditions facilitating the evolution of self-compatibility.

[1] Department of Evolutionary Biology, Uppsala University, Norbyvägen 18D, Uppsala SE-752 36, Sweden. [2] Division of Evolutionary Biology, Faculty of Biology, Ludwig-Maximilians-Universität München, Grosshaderner Strasse 2, Planegg-Martinsried 82152, Germany. [3] Department of Medical Biochemistry and Microbiology (IMBIM), Science for Life Laboratory, Uppsala University, Box 582, Uppsala SE-751 23, Sweden. [4] School of Biological Sciences, University of East Anglia, Norwich Research Park, Norwich NR4 7TJ, UK. These authors contributed equally: Bart P.S. Nieuwenhuis, Sergio Tusso. These authors jointly supervised this work: Jochen B.W. Wolf, Simone Immler. Correspondence and requests for materials should be addressed to B.P.S.N. (email: nieuwenhuis@bio.lmu.de) or to S.I. (email: S.Immler@uea.ac.uk)

Sexual reproduction in eukaryotes generally requires the fusion of two gametes from genetically different, compatible mating partners. The mechanisms of mate compatibility are strikingly diverse across taxa and often involve sophisticated genomic structures ranging from sex chromosomes to highly complex mating types[1–4]. Sexual systems requiring such mate compatibility reduce the probability of encountering a suitable mating partner by up to 50%, when two sexes or two mating types exist[5,6]. Particularly when population densities are low or spatially structured, individual and population fitness may be substantially reduced[7]. The possible evolutionary responses to such selective pressures are (self-compatible) hermaphroditism in animals or self-compatibility in plants, uniting compatible gametes in a single individual[1,7–9].

In microorganisms, and particularly in fungi, mate compatibility is regulated by genetically defined mating types that act at the haploid level[2]. Only when the genes at the mating-type locus (*mat*)—or multiple loci in some groups[10]—differ between the two individuals, in conjugation with successful reproduction possible[11]. Similar to the presence of separate sexes and self-incompatibility systems, the occurrence of mating types strongly limits the chance of finding a compatible mate[5,6]. Even though most fungi are able to reproduce asexually, sexual reproduction is often an obligatory phase of the lifecycle requiring conjugation[12]. Finding a compatible mate is thus crucial for survival[12,13]. Analogous to the loss of self-incompatibility in angiosperms, or the evolution of hermaphroditism in animals and plants[9,14], many fungal species evolved the ability to reproduce without the need of another individual, a state known as homothallism[15,16]. In homothallic organisms, an individual is compatible with all others, including itself and thus can perform haploid intra-clonal mating, while maintaining the potential to outcross. Homothallism evolved several times independently in fungi[2,16], often by combining the genes of both mating types into a single haploid genome[17]. Note that reproductive assurance is not necessarily the only possible explanation for the evolution of homothallism[18]. Alternative reasons may be reduction in outbreeding depression[19] or an increase in compatibility to facilitate outcrossing[2], and these three explanations are not mutually exclusive[13,20]. The automatic advantage of selfing, under which all female gametes are self-fertilized, while additional fertilization by outcrossing through male gametes continues[7,21], might apply to anisogametic, multicellular fungi. In single-celled haploid organisms with isogametic gametes such as yeasts, the contribution of automatic advantage for the evolution of homothallism can be excluded[21].

A specific form of homothallism is mating-type switching, which evolved multiple times independently in both single-celled (i.e. yeasts) and multicellular fungi[18,22,23]. Mating-type switching is a mechanism at the genomic level occurring during mitotic asexual reproduction, which renders the resulting daughter cells sexually compatible with each other. Each haploid cell contains genetic information of both the mating types located at separate mating-type cassettes, but expresses only the mating-type genes at the active locus (*mat1*), whilst suppressing the genes located at the alternative mating-type cassette or cassettes[22,24] (Fig. 1a). During switching, the information at *mat1* is efficiently replaced with the alternative type by a variety of mechanisms[24–29]. While the (epi-)genetic mechanisms of mating-type switching are understood in much detail, little is known about its evolutionary origin and the forces that drive the selection of self-compatibility at the population level[18,22,27]. The two main hypotheses for the evolution of mating-type switching are (i) the lonely spore hypothesis and (ii) the mating-type ratio restoration hypothesis, both suggesting such an evolutionary response to be an adaptation for reproductive assurance[18]. The first hypothesis assumes that dispersal occurs by a single spore, which most often will be alone in the local patch and thus requires intra-clonal mating to complete the sexual phase of the life cycle[22,27]. The latter suggests that when asexual reproduction continues for many generations with repeated bottlenecks or in small populations, the drift will skew the mating-type ratio, which can select for switching to restore this ratio[30]. Asexually produced yeast cells are sensitive to external stress and not expected to survive for prolonged periods in unfavourable environments[31–33]. Repeated bottlenecks of asexually reproducing populations are thus probably not common, favouring the lonely spore hypothesis[22].

Even though many yeast species generally exhibit the ability to switch, switching and non-switching phenotypes can be isolated from nature and these phenotypes may vary between populations. In the budding yeast *Saccharomyces cerevisiae*, for example, 69–87% of the natural isolates are able to switch[34] and in a worldwide sample of the fission yeast *Schizosaccharomyces pombe*, about 85% can switch[35]. Even though laboratory experiments have shown that mating-type switching in the fission yeast can be readily lost (about once per $10^5$ cell divisions[29]), the proportion of non-switchers in nature is multiple orders of magnitude higher than the mutation rate of loss of switching. Here we investigated the molecular, evolutionary and ecological underpinnings for the emergence and persistence of switching vs. non-switching strategies, both in natural populations and in experimentally selected laboratory strains of the fission yeast. We show that mating-type switching is highly beneficial under sexual reproduction, but is costly under asexual growth, which can lead to repeated loss and reappearance of mating-type switching. These findings highlight the importance of understanding the ecological factors for evolution and maintenance of the mixed mating systems in nature.

## Results

**Natural variation of mating-type switching genotypes**. To assess the genetic diversity of structural switching genotypes, we re-analysed the sequencing data from 57 differentiated clades in the worldwide sample[35] (Supplementary Table 1) and observed substantial genotypic diversity at the mating-type region (Fig. 1b and Supplementary Fig. 1). Analyses leveraging the information from sequence coverage provide evidence for distinct genotypes of the mating-type locus with deletions or duplications of several regions, including the Minus (M) and Plus (P) mating-type cassettes (up to three copies for the M and up to five copies for the P cassette), the L region and the K region[29]. The coverage data, combined with mapping the reads to artificially generated structural mating-type variants known[29] to occur in the standard laboratory strain $h^{90}$, suggest multiple novel structural variants. The strains differing in genotype at the mating-type locus showed substantial variation in the ability of phenotypic switching. We analysed the strains described in Jeffares et al.[35] by iodine staining, PCR amplification of the *mat1* locus and test crosses to obtain information for all 57 strains, and found some inconsistencies with previously described observations (see the first and second column in Fig. 1b, Supplementary Table 1 [35,36]). Note that even though genetic changes at other mating-type loci can occur[29], we refer to switching as the change of the expressed cassette at *mat1* renders the strain phenotypically homothallic, i.e. able to perform intra-clonal mating. To explain the evolutionary origin of this diversity, we reconstructed phylogenetic relationships among strains from genetic variations segregating at the regions flanking the silent mating-type region (Fig. 1b and Supplementary Figs 2, 3). The analysis was limited to these regions to extract phylogenetic signals relevant to the loss and potential gain of switching, which might otherwise be confounded by introgression events unrelated to the mating type[35,37].

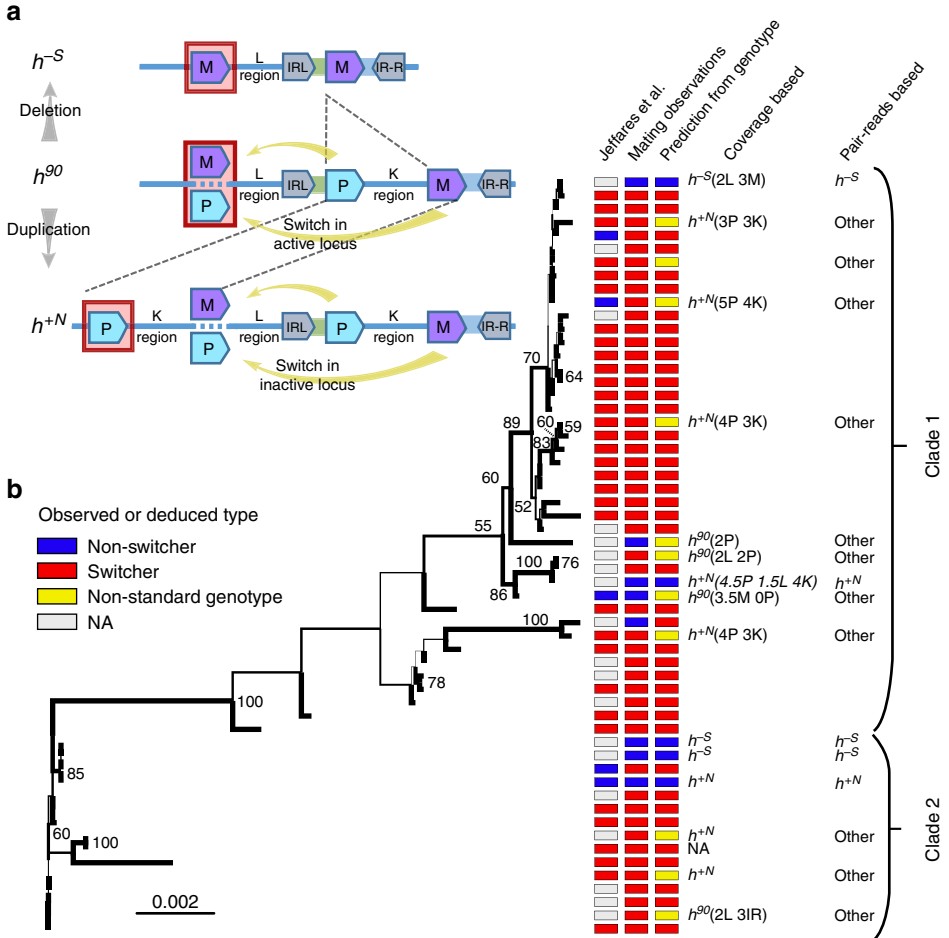

**Fig. 1** The genomic organization of the mating-type region in wild strains and world-wide diversity of switchers and non-switchers. **a** Schematic of the genetic arrangement for switcher ($h^{90}$; middle) and the two most common non-switcher phenotypes ($h^{-S}$ and $h^{+N}$; top and bottom, respectively). Regions are not to scale. The switcher's active mating-type locus *mat1* (red box) switches between Plus (P) and Minus (M), using the silent cassettes as templates during cell division. The non-switcher strain $h^{-S}$ contains only the minus cassette, due to deletion of part of the silent region. The non-switcher strain $h^{+N}$ is the result of a duplication of almost the entire silent mating type and only expresses the P cassette at *mat1* (see ref.[9]). **b** The unrooted maximum-likelihood tree for 57 natural fission yeast isolates, based on the L region (results for the 3′ and 5′ flanking region of entire mating-type region in Supplementary Fig. 3), results in two main clades (indicated by brackets). Branch values show bootstrap support when >50. Colour codes indicate switching phenotypes based on (i) previous observations[35], (ii) our test of single-cell colony mating (mating observations) or (iii) the prediction from inferred genotype. Genotypes were predicted based on the coverage and paired-read analyses, as shown in the last two columns (only shown when differing from the $h^{90}$ haplotype). When genotypes differ from the standard configurations (**a**), additional information on predicted number of copies for the L- and K-region, and P- and M-cassette (Supplementary Fig. 1) is presented between brackets. The samples that do not fit any existing genotypes in **b** are labelled as 'other'

Superimposing phenotypic and genotypic data on the cladogram suggests an ancestral switching state with multiple independent loss-of-switching phenotypes, resulting in unique and several previously undescribed mating-type locus haplotypes (Supplementary Fig. 2).

**Experimental evolution of mating-type switching**. In a next step, we tested whether a switching strategy could also evolve de novo from a non-switching state. To address this question, we conducted an experimental evolution study applying the selection for sexual reproduction under low densities in self-incompatible strains. We created 20 replicate populations from a mix of stable non-switching parental strains of the fission yeast in equal ratios ($h^{-S}$ and $h^{+S}$; see Supplementary Table 1 and Methods for details).These strains lack most of the silenced region including the cassette of the opposite mating type (Fig. 2b)[38] and the K region, which is involved in suppression of the silent cassettes in wild-type $h^{90}$-switcher strains[39]. During one cycle, the strains

were grown asexually on an average of 6.6 mitotic generations, after which sexual reproduction was induced by transferring 1% of the cells to solid low-nitrogen medium (Fig. 2a). After the sexual cycle, 1% of the cells were harvested, of which only sexually produced offspring (i.e. the ascospores) were maintained by killing all non-mated cells using Glusulase (a digestive enzyme mixture) and subsequent 30% ethanol treatment. These ascospores were then transferred to a fresh growth medium to start the next cycle. In contrast to the diplontic-budding yeast[40,41], the fission yeast is haplontic and no special measures are necessary to avoid fusion during asexual growth. After 25 serial cycles, nine of the 20 evolved replicate populations consisted of 15% to 100% of genotypes that were able to mate within single colonies (Supplementary Fig. 4a). In other words, the selection for sexual reproduction at low densities resulted in the evolution of the ability to produce sexual spores from a single haploid individual in multiple populations. Haploid sporulation[42] and diploidy[43] could be excluded as a cause of sporulation (Supplementary Fig. 4b, c). PCR for the *mat1* locus containing the active cassette

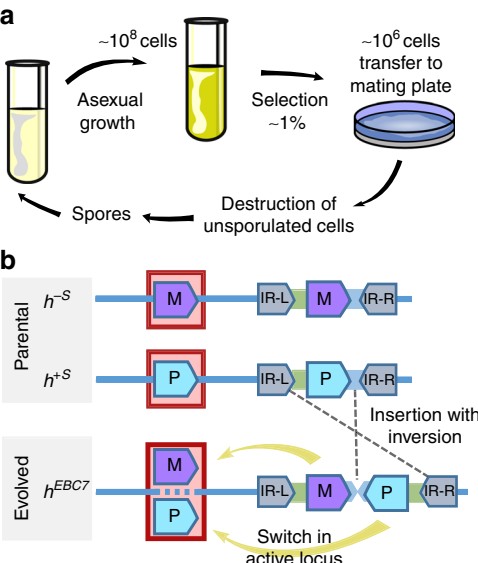

**Fig. 2** Experimental evolution of switching. **a** Sequence of events during one experimental cycle: (1) 2 days of asexual growth to saturation. (2) Transfer of 1% of the cells to a nitrogen-deficient medium. (3) Killing of non-mated haploid and unsporulated diploid cells with enzyme treatment and ethanol. (4) Transfer of sexual spores to a fresh medium for asexual growth. **b** Mating-type configuration of parental strains used in the experiment (top) with non-switching phenotype ($h^{-S}$ and $h^{+S}$), and configuration of the long-read sequenced-evolved switching strain (EBC7; bottom) with an active mat1 locus indicated by the red box. The evolved strain contains an additional mating-type cassette, which allows the strain to switch between P and M

showed the presence of both M and P cassettes, providing further evidence for functional switching (Supplementary Fig. 4d). As a proof of principle to understand the genetic mechanism by which homothallism was restored, we sequenced one of the evolved switching strains (EBC7) by whole genome long-read SMRT sequencing (Pacific Bioscience, CA), which confirmed the incorporation of both cassettes into the silenced locus. However, in contrast to the wild-type genotype of the switching $h^{90}$ strain, the M cassette was positioned left (i.e. centromere proximal) from the P cassette. In addition, the P cassette was inverted with the H1-regions of the silent P and M cassettes close to each other (Fig. 2b and Supplementary Fig. 5), resembling the arrangement in the two related fission yeast species for which the mating-type structure is known, viz. S. japonicus and S. octosporus[44]. To assess the structure of the silent mating-type regions in other strains, we performed PCR using primers on the left or right of the inverted repeats (IR-L and IR-R; Fig. 2b) at the edge of the silent region, which we combined with P- and M-specific primers. The PCR results showed three different mating-type configurations (Supplementary Fig. 6); the most common configuration suggested that the entire silent cassette including both inverted repeats was duplicated; once for P and once for M, with a duplication of (part of) the L region.

These findings suggest that mating-type switching can readily evolve as a response to selection. In just 25 generations, a mutation occurred that increased to significant levels in five populations and was fixed in four populations (Supplementary Fig. 4a). Even though the mating-type region is known to be unstable and prone to meiotic recombination and gene conversions[45,46], we estimate the spontaneous mutation rate of the mating-type switchers in the fission yeast laboratory strains to be relatively low. We screened a million spores for self-compatibility

using iodine vapour staining, which showed no spontaneous switchers arising after meiosis (see Methods). This corresponds to an upper limit of the mutation rate, $\mu$ of $3 \times 10^{-6}$, under binomial sampling ($P_{(K=0; N=10e6; p=1 \times 10e-06)} < 0.05$), which is clearly not sufficient to explain the recurrent fixation in nine populations. Note that the kind of structural mutations we observed here is unlikely to spontaneously appear during mitotic cell division, but is instead expected to occur through recombination during meiosis. Genotyping of 50 individuals from the evolved population that was sequenced showed that all sampled individuals had the same inversion as the strain EBC7, suggesting a single mutation that rapidly got fixed. The occurrence of mutations generating a mating-type switching phenotype within 25 sexual generations, in 9 of our experimental populations kept under low population densities and (near) its fixation in 7 of them, indicates that such mutations are efficiently selected for under these conditions. Assuming strong selection (see below for justification of this assumption) and a population size of 1.25 million zygotes (i.e. 2.5 million potential crossing overs that can result in a switching recombinant) per generation, we can make a rough estimate of the lower limit of the mutation rate, which is nine mutations in 25 generations × 20 populations × 2.5 million $= 7.2 \times 10^{-9}$ mutations per sexual generation. Note that this is a conservative estimate, as it ignores drift during the asexual growth phase. A novel switching genotype might be lost when 1% of the population is subsampled after growth with a chance of $P_{(K=0; N=5 \times 10e08; p=5 \times 10e-08)} = 0.37$ under binomial sampling, resulting in a final estimate of $1.14 \times 10^{-8}$ mutations per sexual generation. This estimate lies far above the per generation mutation rate for mitotic division of $10^{-10}$[47], but is much lower than the spontaneous rate of reversion from the non-switching $h^{+N}$ genotype to the switching of $9 \times 10^{-5}$ mutations per mitosis[29] and our upper limit of $3 \times 10^{-6}$ mutations per meiosis. Overall, this suggests that repeated evolution of switching is not the result of a mutation bias, but caused by strong selection under relatively low mutational input. The exact rate will likely be dependent on the genotype of the mating locus.

**Quantifying the benefits and costs of mating-type switching.** The remarkable ease with which the novel mating-type switching genotypes arose and could be selected for under experimental conditions suggests that this trait can, in principle, readily emerge and persist in nature. Little is known about the natural habitat of the fission yeast[48], but most likely growth occurs in a static structured environment[36], similar to budding yeasts[32]. Due to its haplontic lifecycle, a non-switching fission yeast genotype that dispersed to a novel location can only reproduce sexually if another strain is present. After local asexual growth, the cell clones will be clustered and unable to self-fertilize and sporulate in the absence of a switching mechanism. Inability to form spores is likely detrimental, because even though strains can disperse as vegetative cells, these cells are fragile and prone to aging[49], while sexual spores are highly resistant to external conditions and can remain dormant for long periods of time[33].

We experimentally quantified the benefits of the switching phenotype during sexual reproduction by competing unevolved isogenic switching and non-switching lab strains with each other under variable densities using fluorescent labels to distinguish between the strains. We mixed the cells in standardized proportions of 50% switchers, 25% non-switching P and 25% non-switching M and tested the competitiveness of the switchers vs. the non-switchers under four different densities ($8.0 \times 10^5$ cells mm$^{-2}$ with three 10-fold serial dilutions). The highest density is approximately ten times higher and the lowest dilution slightly lower than the density at saturation in liquid medium used during

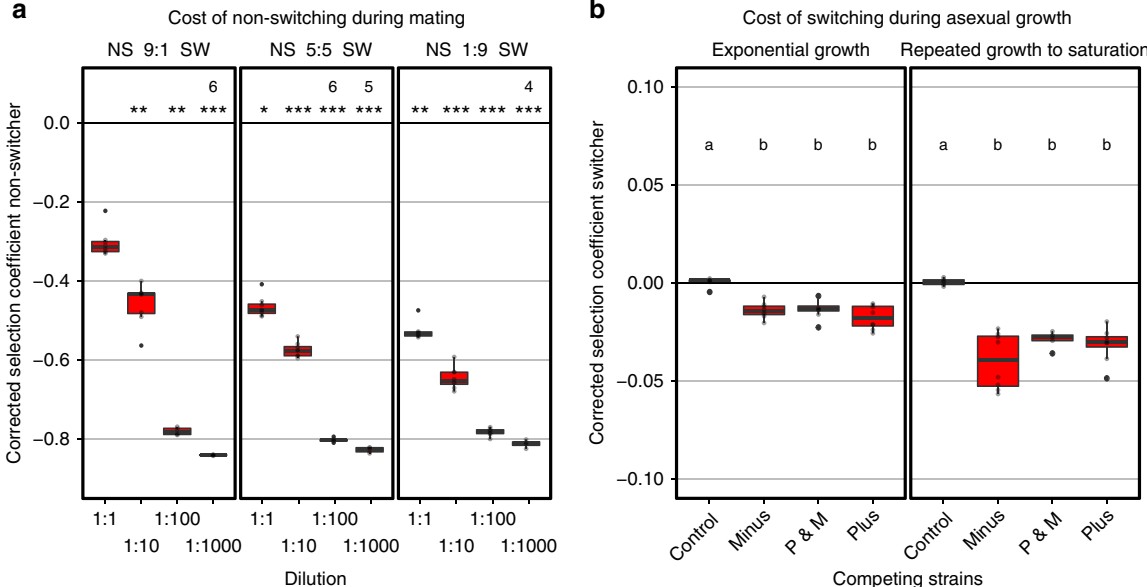

**Fig. 3** Fitness benefits and costs of mating-type switching (MTS) during sexual and asexual reproduction. **a** Selection coefficients for non-switchers (NS), relative to switchers (SW), after one round of sexual reproduction under four different densities in a structured environment ($8.0 \times 10^5$ cells mm$^{-2}$ in highest density) on inverted $y$-axis to indicate reduction in fitness. Non-switchers decrease in fitness at low densities and the benefit of switching is the highest when switchers are at high frequency in the population, regardless of density. The boxplots are based on eight biological replicates (open circles; exceptions are indicated by number above box; see Methods). All strains competing are isogenic, except for mCherry marker and switching locus. Asterisks indicates significant deviation from the control (Dunn's Kruskal-Wallis multiple comparison, $*p < 0.05$, $**p < 0.01$, $***p < 0.001$). **b** Per generation selection coefficient for unlabelled switcher, relative to mCherry labelled non-switchers of M, P or a mixture of equal numbers of P and M, under asexual growth after 5 days of exponential growth (~45 generations; left panel) or after five transfers of growth to saturation (~33 generations; right panel). Control is the competition between the wild-type strain $h^{90}$ (EBC5) and itself but mCherry labelled (EBC47). Each boxplot consists of eight technical replicates (grey dots). Letters within each panel indicate significantly different groups based on Dunn's Kruskal–Wallis multiple comparison ($p < 0.05$). Note inverted $y$-axis

experimental evolution. These mixes were incubated for 3 days on low-N medium to induce mating, after which the non-sporulated cells were destroyed and the spores were germinated to obtain sexual offspring. We then compared the frequencies before and after competition using flow cytometry and controlling the potential effects of the fluorescent marker on the results. In line with our predictions, the frequencies of switchers generally increased under all densities, but significantly more so at lower densities (Supplementary Fig. 7). Non-switchers were under all circumstances selected against depending on the density regime. The selection against non-switchers was the weakest in high density regime (undiluted; $s = 0.303$) and strongest under the lowest density regime (1:1000 dilution), with the selection coefficients reaching 0.837 (Fig. 3a and Supplementary Fig. 8). We assume this is an underestimate of the actual selection coefficient, because of a high background noise in the results at the highest dilution (see Methods). To test for frequency dependency, we competed the switchers and the non-switchers at different ratios (9:1 and 1:9) and showing positive frequency dependence under high densities (Supplementary Fig. 9). Only when the non-switchers were present at high densities as well as at high frequencies relative to switchers did they perform equally well as the switchers. Mating-type switching is thus not only beneficial with respect to mating-type dependent compatibility, which is always balanced irrespective of the switcher/non-switcher ratio, but is intrinsically advantageous. Two switched and now compatible daughter cells will have synchronized cell cycles required for mating initiation[11] that are thus no longer available as potential mates for non-switchers. Only at very high densities and frequencies will the non-switchers have a high chance to find another cell of the opposite mating type that is

unmated, and therefore likely non-switch. Non-switchers are thus unlikely to invade a sexually reproducing population of switching cells.

Given the apparent benefits of the mating-type switching strategy, the stable existence of strains with non-switching phenotypes in nature suggests that switching may involve costs. One possibility is that the switchers are inferior competitors, relative to non-switchers, during asexual growth, as suggested by the fact that most non-switching natural isolates are derived from human-associated nutrient-rich environments where prolonged asexual propagation is likely[35,36,48] to occur. We measured the potential costs associated with the mating-type switching phenotype empirically by competing the switchers with the non-switchers, marked with different fluorescent labels during asexual growth. An equal number of switchers and non-switchers was mixed and propagated under two growth regimes—to test for generality of results under two highly distinct cell-cycle dynamics —by transferring 1% of cells to a fresh medium (i) every day (maintaining continuous exponential growth) or (ii) every second day (which allows growth to saturation before each transfer). Comparing the frequencies of the switchers and the non-switchers by flow cytometry before and after competition showed that switchers grew at a slower rate than non-switchers (Kruskal–Wallis; exponential: $H = 16.79$, df $= 3$, $p < 0.001$; saturated: $H = 16.49$, df $= 3$, $p < 0.001$). This translates to a selection coefficient per cell division of (i) 0.016 (StErr 0.00022) for the continuous growth regime and (ii) 0.035 (StErr 0.00053) in cycles with growth to saturation (Fig. 3b and Supplementary Fig. 10), indicating a potential cost of phenotype switching under prolonged asexual growth under two different regimes.

## Discussion

Both competition assays and an evolutionary experiment show that mating-type switching in the fission yeast has a large selective benefit under scenarios where regular sexual reproduction is required. However, this strategy is also associated with costs under prolonged asexual growth, which could explain the occurrence of both switching and non-switching strategies in natural populations. In a natural environment, a typical lifecycle involves dispersal, followed by a short burst of asexual growth associated with resource-depletion-induced sporulation producing dormant and resistant haploid spores[36]. The production of spores is critical for yeasts, because mitotic haploid cells are sensitive to many forms of stress (e.g. heat, desiccation and digestion), to which the meiotic spores are resistant[33,50]. In most fungi, production of resistant cells is completely linked to the sexual cycle and thus completely depends on successful mating[12,51]. In the here-assumed lifecycle of the fission yeast, the cost of switching, relative to non-switching, during mitotic growth will be minimal, because the number of cell divisions is low, while the mating opportunity is optimized for equal and well-mixed mating-type ratios. Understanding the evolution of non-switching strategies in nature requires insight into the conditions that make mating-type switching costly when such conditions are met.

We hypothesize that the cost of switching documented here is likely to involve several mechanisms. Firstly, asexual growth is probably slowed down by the switching mechanism itself. Switching is initiated during DNA synthesis by an epigenetic mark at the *mat1* locus—possibly a single-strand nick[52] or RNA residue[53]—which requires repair that is associated with a lag in DNA synthesis[24]. However, even when no switching occurs, the epigenetic mark and associated repair are maintained in non-switching strains[54,55] and hence, this process cannot account for

the entire cost. A second possible cost is pheromone-induced mitotic arrest stalling cell cycle at G1 phase[56]. Each mating type produces its own pheromone, which induces arrest in cells of the opposite mating type. In switching strains, both pheromones are produced, inducing a consistent delay in cell cycle. In contrast, non-switchers produce only one pheromone to which they themselves do not respond. When a skew in the mating-type ratio exists, a positive frequency dependence can be observed in competitions between non-switchers of opposite mating types, favouring mating type at the highest frequency (Supplementary Fig. 11). However, frequency dependence cannot explain the observed reduced fitness for switchers, which also occurred under balanced mating-type ratios, which should reduce growth of both types equally. We think the most likely cost of switching is associated with the change in the phenotypic cell identity. True homothallic fungi express both mating types at the same moment, which is assumed to be costly due to unwanted interactions between mating types. Mating-type switching might have evolved to overcome these costs of expressing both at the same time by separating their expression in the daughter cells[18]. However, the variety of mating-type-specific proteins expressed in the cell is equally distributed among the two daughter cells during mitosis. Expression of the new mating-type identity, while retaining proteins from the previous identity, can lead to autocrine reactions that are likely to interfere with the mitotic cycle[57].

To shed further light on the dynamics of mating-type switching in a structured environment, we performed individual-based cellular-automata simulations (see Methods for details). In short, a grid of $200 \times 200$ cells was populated at different densities with switcher and non-switcher individuals of either mating type, which grow asexually by occupying an adjacent grid cell until the grid is completely occupied. Once occupation is completed,

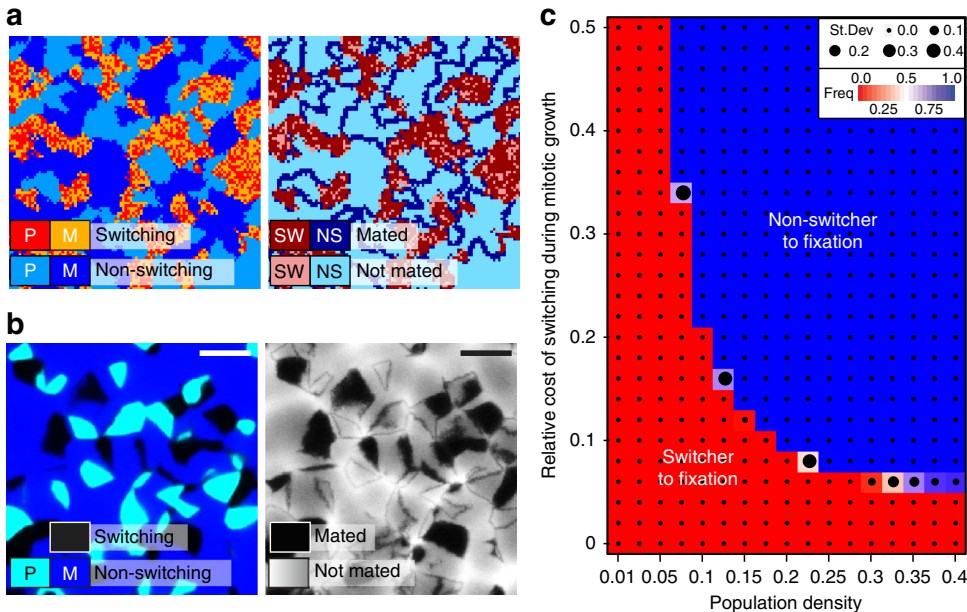

**Fig. 4** Growth in structured environments greatly benefits switchers during mating. **a** Example of simulation. Left panel shows result of one round of asexual growth of simulation that shows the mating type of switcher (SW) (red and orange for P and M, respectively) and non-switchers (NS) (light and dark blue). The right panel shows, which cells mated and contributed to the next generation. **b** Left panel shows recoloured composite fluorescence image of a mixture between two labelled non-switching P and M strains (NS; two shades of blue; mCherry and GFP, respectively) and an unlabelled switching strain (SW; no fluorescence, i.e. black) plated out at low density on low-N medium. Right panel shows the same plate after iodine vapour was applied to stain mated cells (black). Switcher patches are fully mated, non-switchers can only mate at the edge of a patch similar to simulations in **a**. Scale bar: 0.5 mm. **c** Heat map summarizing the results from cellular automata simulations in a 40,000-cell grid after 500 transfers or when either switchers or non-switchers went extinct, with varying levels of population density and cost of switching. Under low densities or with little cost of switching, non-switchers cannot invade in the population. Stable polymorphism is not observed. Colour indicates average ratio of non-switchers from 10 replicate runs with standard deviation indicated by the size of the black dot

mating occurs between compatible neighbours, and the randomly selected sexual offspring are used to initiate a next cycle. Our simulations showed that while switchers were able to mate across their clone-patch, non-switchers were only able to mate at the border of the colony where they encountered cells of the opposite mating type ('contact zone', see Fig. 4a). These differences in patterns were particularly pronounced under low density, which increases the patch size, relative to the contact zone length (Supplementary Fig. 12a–d). The emergence of patterns differing by mating type was corroborated in vivo using switchers and non-switchers with different fluorescent labels (Fig. 4b and Supplementary Fig. 12e–g). These findings confirm our hypothesis that switching may be particularly beneficial in a structured environment. Adding a cost to mating-type switching during asexual growth in our simulations allows non-switchers to invade a switcher population, but invasion only occurs when both, the cost of switching and cell densities, are high enough (Fig. 4c). Incorporating a cost to the next generation for genotypes created from intra-clonal selfing into our simulations does not change the result of the simulations. Even though this is not equivalent to the accumulation of deleterious mutations by haploid selfing, switching should not greatly increase inbreeding depression, because it does not reduce the potential for outcrossing and can hence reduce the cost of inbreeding. Interestingly, outcrossing rates in the fission yeast in nature, based on linkage disequilibrium, is estimated to occur only once per 600,000 generations[47], which is substantially lower than the instance in the budding yeast (once every 100,000 asexual generations)[58]. Potentially, due to S. pombe being haplontic, deleterious mutations are not sheltered and thus visible to selection, which might reduce selection for outcrossing[41]. The low rate of outcrossing furthermore suggests that sexual reproduction occurs mostly for purposes other than recombination, most likely to generate spores for survival. The fact that no yeasts have evolved euploid survival propagules[42] without recombination suggests that sexual reproduction is an indispensable component of spore formation[51].

The cost of mating-type switching during asexual reproduction (~1.5–3.5% per mitotic generation) should lead to the loss of mating-type switching under conditions favouring asexual reproduction. With an estimated effective population size of 12 million[47] and a mutation rate towards non-switching of $10^{-5}$ [29]), this would suggest that a non-switching mutant should go to fixation in approximately 460 to 1100 generations[59]. The observed frequency of 15% non-switchers suggests that sexual reproduction, which strongly selects against loss of switching, should occur more regularly to avoid the loss of switchers. Regular sexual reproduction greatly slows down the fixation of non-switching, even when the costs of switching and densities are very high (Supplementary Fig. 13). This estimated rate of sexual reproduction is higher than the expected frequency of meiotic reproduction in the budding yeast, S. paradoxus (once per 1000 asexual generations)[58], which is induced to sporulation by similar environmental cues, as is the fission yeast. The estimated selection coefficient of about 0.82 per sexual generation at the lowest concentration, which we expect to be still much higher than under natural densities[48], should outweigh the costs of switching during asexual growth to explain the maintenance of mating-type switching.

Alternatively, reversion or de novo re-evolution of a switching genotype from two non-switchers might help maintain mating-type switching. Rearrangements during meiosis are common[29], and together with strong selection for reinstatement of switching may be responsible for the variation at the mating-type region we observed (Supplementary Fig. 1). This is even possible when locally only one mating type occurs, because in S. pombe a variety of heterothallic genotypes spontaneously revert to switching[29].

Summarizing, our data predict that non-switcher strains isolated from nature should have evolved either under prolonged asexual reproduction, in high-density environments or in locations where long-term survival is not essential, for example, when resources are regularly replenished. Similarly, the costs during asexual growth might also explain the strongly reduced frequency of mating-type switching (e.g. C. glabrata[60]) or even the complete loss of switching in opportunistic pathogenic fungi (e.g. Lachancea kluyveri[61]) that mainly grow asexually and where the immune system is likely to impose strong short-term selection on locally adapted strains[62]. An alternative to reducing the costs is to switch only when mating is required, as occurs in some methylotrophic yeast where switching is induced by the same cue as mating[27].

Mating systems are highly variable and many transitions towards self-compatibility, such as hermaphroditism and the loss of self-incompatibility, have been shown in a variety of taxonomic groups[1,9]. The recurrent transitions between such mating systems and their co-occurrence in natural populations[63] indicate little phylogenetic constraint allowing for rapid, context-dependent shifts. Our study provides evidence for the de novo evolution of self-compatibility by means of mating-type switching and quantifies the ecological parameters that govern context dependence. The switching phenotype showed significant evolutionary advantage—especially under low densities when mate availability is limited—but incurred increased cost during asexual growth. Theoretical simulations support the hypothesis that this trade-off between mating system and growth efficiency could explain several independent transitions among natural populations. Our results emphasize the importance of understanding the ecological factors that dictate these costs and benefits to predict the repeated evolutionary transitions to and from self-compatibility.

## Methods

**Natural diversity in mating region and switching ability**. We used published data from 57 world-wide distributed strains (description of samples in Jeffares et al.[35]) to build phylogenetic trees of sequences flanking the mating-type region. We used VCF files with SNP information for each individual and extracted the genetic variants within 10 kb of the sequence flanking the mating region and 9.8 kb of the 13.8 kb long L region (2 kb were excluded on each side to reduce possible linkage with mating loci). Both, left and right flanking region as well as the L region were analysed independently. The same analysis was conducted using 5 kb on each side with qualitatively similar results. Fasta alignments of the 57 samples were generated by substituting SNP variants into the reference S. pombe 972 $h^-$ reference genome (ASM294v2[64]) using the package vcf2fasta (https://github.com/JoseBlanca/vcf2fasta/). Maximum likelihood trees were inferred with RaxML (version 8.0.20[65]) using default parameters, GTRCAT approximation, final optimization with GTR + GAMMA and 1000 bootstraps.

**Genomic inference from coverage**. The published raw resequencing pair-reads from the same 57 clonal strains[35] were used to infer genetic structure of the mating region. All samples were sequenced in triplicate to an average of 76-fold coverage, with a minimum of 18-fold coverage in 42 out of the 57 samples. Adapters were removed using the package cutadapt (version 1.9.1), and the reads were cleaned with fastq-mcf from the package ea-utils (version 1.1.2). Reads were mapped to the mating-type region of the $h^{90}$ reference ('MTR chromosome' in ASM294v2 assembly[64]) using BWA (version 0.7.13) with default settings. The package Samtools (version 0.1.19) was used to remove PCR duplicates and to filter by duplicate reads and mapping quality. GATK (version 3.6) was used to extract coverage information per base. The mean coverage per base was estimated for each locus or genomic region in the mating-type region (Fig. 1a) and normalized by dividing with the mean coverage per base in the two flanking regions. For the flanking regions, 5 kb were used left of mat1 and right of the silent region (referred to as 'left' and 'right'), excluding 1 kb adjacent to the MT region. To reduce mapping conflict between homologous sequences, the genomic regions between 300 and 1000 bp at the beginning and end of each genomic region were excluded from the analyses. Within the K region, the cenH locus was excluded due to sequence similarity with centromeres. The normalized coverage per base was used to infer the number of duplicate sequences for each locus or region.

**Pair-read inference of genotype**. The structural organization of the mating-type region is difficult to assemble de novo using short-read re-sequencing data, given

the presence of regions with copy number variation. We tested the configuration at the *mat1* locus by in silico mapping the raw Illumina reads from Jeffares et al.[35] to artificial scaffolds of known mating-type region arrangements focusing on the regions that can potentially flank a mating-type cassette on either side. The sequences used for construction of these scaffolds were extracted from the reference sequence (ASM294v2[64]), in which the MTR region from the $h^{90}$ haplotype was inserted at the *mat2:3* locus, as described in http://www.pombase.org/status/mating-type-region. In total, 12 scaffolds were built (details in Supplementary Data 1 and Supplementary Fig. 14). All the scaffolds were compiled in a fasta file and used as reference sequences for mapping. The read pairs were mapped using BWA (version 0.7.13) and Bowtie2 (version 0.12.6) with default settings. The paired reads, in which each read mapped to a different genetic region or locus within a scaffold, were filtered out. The final number of the mapped pair reads were used as a measure of support for different structural genetic arrangements (Supplementary Data 1). For samples that were sequenced multiple times, each replicate was analysed independently combining the results for the overall genetic inference.

**Experimental evolution**. Experimental evolution was performed using $h^{+S}$ and $h^{-S}$ strains derived from Leupold's 968 strain[38]. Twenty replicate populations were maintained: ten, where both strains carried the *ade6-M216* allele (EBC1 and EBC3) and ten, where both strains carried the *ade6-M210* allele (EBC2 and EBC4). We used standard liquid Edinburgh Minimal Medium (EMM; Per liter: Potassium Hydrogen Phthalate 3.0 g, Na HPO$_4$·2H$_2$O 2.76 g, NH$_4$Cl 5.0 g, D-glucose 20 g, MgCl$_2$·6H$_2$O 1.05 g, CaCl$_2$·2H$_2$O 14.7 mg, KCl 1 g, Na$_2$SO$_4$ 40 mg, Vitamin Stock ×1000 1.0 ml, Mineral Stock ×10,000 0.1 ml[66]) for the asexual growth phase and 2% agar solid Pombe Minimal Glutamate medium (PMG; EMM in which ammonium chloride is replaced by 5 g l$^{-1}$ glutamic acid) for the sexual growth phase. Both were supplemented with 100 mg l$^{-1}$ adenine.

The cells were maintained in 30 ml tubes containing 5 ml EMM at 32 °C and shaking at 250 rpm. After 48 h, the medium was saturated (~$10^8$ cells per ml) and a subsample containing 1% of the cells (around $5 \times 10^6$ cells) were transferred to a fresh 4.5 cm plate of PMG. The cells were placed in the middle of the plate and allowed to spread out. The resulting density is around $3.1 \times 10^3$ cells per mm$^2$, where a cell of 40 μm$^2$ needs to go on average through ~4 cell divisions until one of its clones meets another colony. After 3 days of incubation at 25 °C, the cells were collected from the plate by drawing a toothpick across the patch of cells. The collected cells were resuspended in 1 ml sterile MQ water containing 15 μl Glusulase (PerkinElmer, Waltham, MA) and maintained at 25 °C overnight, which destroyed the cell walls of the most haploid and diploid cells and the ascus wall, but did not harm the ascospores. In a next step, 96% ethanol was added, resulting in a final concentration of EtOH 30% (v/v), and the suspension was incubated for 30 min at room temperature to kill the remaining haploid and diploid cells without affecting spore survival. Tests showed the survival of non-sporulated cells of <0.1%. The ascospores were pelleted, the supernatant was removed and the ascospores were resuspended in a fresh 5 ml EMM for the next cycle. After 25 cycles, a random sample of the cells was spread on 9 cm PMG plates (~500 cells per plate) and incubated for 5 days at 25 °C. The formed single-cell colonies were stained with iodine vapours[66,67] and scored for the spore production. Single colonies were transferred multiple times to check for stability of the genotype. We performed multiple tests to exclude the potential mechanisms by which a single cell can reproduce sexually without a partner:

We tested whether cells are diploid, which might explain single-cell colony sporulation. If the cell is diploid before plating out the single colonies, it can divide mitotically multiple times before sporulating. Using Phloxin-B plates, which stain diploid colonies dark pink but haploid ones in light pink[43,66], we only observed haploid colony phenotypes.

Some mutant strains are able to sporulate in the haploid state[68] by sporulating from a single cell without conjugation. We visually inspected >100 asci per strain to verify the shape of the ascus. In all cases, at least half of the asci showed the distinct irregular zygotic ascus phenotype, indicating that sporulation occurred immediately after zygote formation.

To identify the mating type present at the *mat1* locus, we performed colony PCR with primers mat1_fwd as forward and P_rev or M_rev as reverse for *mat1P* and *mat1M*, respectively, with $T_m = 52$ °C (Supplementary Table 2).

We performed DNA extraction for SMRT Sequencing (PacBio®, CA) using the Genomic Tip 500/G kit (Qiagen) following the manufacturer's protocol. Sequencing was performed at SciLifeLab, Uppsala using three SMRT cells at 20 kbp insert size, resulting in an average coverage per base of 212× for the nuclear genome. The genome was assembled using Canu V1.3 with standard parameter settings. The De-novo scaffolds containing the MTR were identified by nblast using Blast 2.2.29+ and annotated using python scripts (https://github.com/situssog/Visual_Scaffolds).

We verified the occurrence of multiple mating-type switching genotypes within a single population. We randomly picked 50 single-spore colonies from the population that contained the strain EBC7 sequenced with the PacBio® Technology and performed PCR (IR-R-right-Rev and mat-plus-R; Supplementary Table 2) to verify presence of the inversion of the silent matP, which occurred uniquely in this population. If another genotype would have evolved in this population it would be likely to be by duplication, as was observed in most other populations.

We estimated the meiotic mutation rate towards switching by analysing many spores for the presence of self-fertility. The non-switching strains EBC1 and EBC3 were inoculated together on PMG for 72 h and afterwards treated with Glusulase and 30% ethanol, as described above. To reduce resampling the same ascus, we analysed 10% of a total of approximately 10 million spores, which corresponds to $8.6 \times 10^5$ independent asci sampled (estimated by simulating 1000 replicate samplings). The spores were resuspended in the yeast extract medium[66], incubated for 6 h at 32 °C to induce germination, but before a first division is finished[69], and were plated out at a density of approximately 10,000 spores per plate on 9 cm Petri dishes with PMG on a total of 100 plates. After incubation for 7 days, the plates were stained with iodine and scored for black colonies that could indicate homothallic strains. All dark colonies, potentially indicating a mutation to homothallism (52 in total), were transferred to fresh plates and streaked out on PMG and incubated for one more week. These plates showed that all dark colonies were actually a mixture of non-switching plus and minus cells, thus none of the 100 scored plates contained mutants with the switching phenotype.

**Fitness measurements asexual growth**. To assess the effect of only the mating system in our fitness measurements, isogenic S. pombe strains are required that differ only for this trait. To obtain these strains, we plated strain EBC5 ($h^{90}$ leu1-32; see Supplementary Table 1 for all strains used and their names) in low density on PMG + leu plates, and using iodine vapour as described above isolated 50 spontaneous mutant non-switching colonies. We assessed the mating type by crossing the strains to known P and M strains. We randomly picked four strains of each mating type (strains EBC37 to EBC44) and performed three PCR (mat2:3-1-F & mat2:3-1-R, mat3:1-1-F & mat3:1-1-R, and mat1l-K-1-F & mat1l-K-1-R; Supplementary Table 2) to identify the mating-type locus configuration. This confirmed that all M strains were $h^{-S}$ and all P were $h^{+N}$ (Fig. 1a).

To distinguish between strains in flow cytometry and imaging analyses, a fluorescent mCherry-marker was introduced by transformation of the strain EBC5 using standard electroporation techniques[66] using plasmid pKS394 with primers Ssm4_right_Rev and Ssm4_right_Fwd, which are directed to a non-functional region on chromosome I to the right of the *Ssm4* gene[70,71]. A single stable transformant (EBC47) was chosen and used to cross the insertion into the non-switching isogenic strains to obtain eight corresponding fluorescent strains (EBC59 to EBC66). In sexual competitions, even though the fluorescent marker is unlinked to the mating-type region, the frequency of the marker remains a direct measure of the mating success of the labelled strain, due to the Mendelian characteristic of the trait.

Asexual fitness measures were performed by mixing the cells (grown overnight in EMM to approximately $10^7$ cells per ml) from fluorescent and wild-type strains, diluting the cells to $10^6$ cells per ml and using this mixture to inoculate the growth medium (EMM; always supplemented with 225 mg l$^{-1}$ leucine in the 'fitness measurements'). The actual frequency of the strains was measured before and after competition with a flow cytometer (BD LSR Fortessa, at BioVis, SciLifeLab Uppsala, Sweden) and the relative fitness was calculated from the change in frequency. To control the potential marker effect, we performed all competitions with the marker in the switcher and in the non-switcher background. In addition, we performed controls between isogenic lines with and without marker (EBC5 vs. EBC47, EBC37 vs. EBC59, etc.) to measure the potential marker effect in each background. Measurements were analysed in R using the package flowCore[72]. Due to unequal variance among groups, statistical significance was measured using a Kruskal–Wallis test with a Dunn's post hoc test for significance.

We calculated the selection coefficient *s* between competing strains per mitotic generation using the difference between the log of the ratios of the frequencies (*p*) of the competing strains $s_{ij} \approx \log(p_{i,\text{after}}/p_{j,\text{after}}) - \log(p_{i,\text{before}}/p_{j,\text{before}})$ for the competing strains *i* or *j*, assuming *s* to be small[73]. The total number of generations *t* was calculated for five transfers with a 100-fold increase of binary fission, i.e. ~6.6 generations/transfer. We furthermore assumed that the selection coefficients are additive and we can thus directly correct the marker effect, $s_{\text{switching}} = s_{\text{total}} - s_{\text{marker}}$. The effect of the marker was measured under equal circumstances and densities as the competitions, using competitions between isogenic strains that differed only in the marker. This assumption fits well with the symmetry observed in competitions with reciprocally marked switching and non-switching strains (see Supplementary Figs 8, 10).

For asexual competitions in which cells were maintained in continuous log phase, cells were grown overnight in 5 ml tubes EMM, diluted to $10^6$ cells per ml EMM and mixed in desired ratios. These mixtures were then inoculated into deep well plates (Eppendorf Plate Deepwell 96/1000) with 300 μl per well and covered with gas permeable rayon film to reduce evaporation differences between the wells. The plates were incubated at 32 °C on a plate shaker at 400 rpm. Every day, 0.2% of cells (i.e. 0.6 μl) were transferred to wells with fresh EMM. Frequencies were measured after five transfers (~45 generations). We performed competitions with eight biological replicates by competing the switcher strain EBC5 with each of the non-switching fluorescent strains (EBC59-66).

For the competitions with repeated growth to saturation, the procedure is similar to the continuous log phase, but transfers were performed every second day, which gives the culture the time to grow to saturation. Frequencies were measured after five transfers (~33 generations).

**Fitness measurements sexual competitions.** The potential benefit of MTS during sexual reproduction was tested by competing switching and non-switching strains (see Fitness measurements asexual growth for strains used) in a similar way, as described for asexual competitions. Cells were grown overnight to exponential phase, concentrated to $10^8$ cells per ml and mixed in the required ratios, which were estimated using flow cytometry (see above). We made serial dilutions (1:10, 1:100, 1:1000) from these mixes and inoculated these at 25 µl per well in 96-well plates with 100 µl solid PMG per well. The plates were air dried and incubated for 3 days at 25 °C. Assuming that the entire surface of the well (~3 175 µm²) will be evenly covered, this means that the cells with an average size of 4 µm by 10 µm would cover the plate with ~10 layers of cells in the highest dilution. At the 10-, 100- and 1000-fold dilutions, a cell has to grow on average 1, 4 or 7 generations, respectively, to reach non-self-cells assuming exponential growth.

Three days after mating, we harvested all cells by adding 100 µl sterile water to each well, covering the plate with foil and placing it on a plate for vortexing at 2000 rpm for 15 min to loosen the cells from the solid medium. The cells were further loosened by gently pipetting the solution ten times after which 50 µl water with the cells were transferred to 30% ethanol for 30 min to kill non-sporulated cells. The cells were pelleted and resuspended in fresh EMM and incubated shaking at 32 °C for 24 h. In the end, the number of fluorescent cells was measured in the flow cytometer. Measurements were analysed in R, using the package flowCore[72]. Each competition was performed with eight technical replicates, because low cell density increased the noise (dead cells and ascus-wall debris) to signal (live cells) ratio in some low-density samples requiring the exclusion of some replicates (see Fig. 4a).

Fitness was calculated for non-switchers, relative to switchers, similar to the measurements described above, $\log(w) = \log(p_{i,\text{after}}/p_{j,\text{after}}) - \log(p_{i,\text{before}}/p_{j,\text{before}})$ and the selection coefficient $s = 1 - w$[73]. Again, we corrected for the marker effect assuming the selection for switching and the marker to be additive. Note that a difference between competitors can arise from growth before mating, mating success and growth after mating. The difference, relative to the control, for the respective dilution was tested using a Kruskal–Wallis test for significance with a Dunn's post hoc test.

**Cellular automaton simulations.** We performed individual-based simulations using a custom script in R 3.2.6, using the package simecol (http://simecol.r-forge.r-project.org/). A finite grid of 200 × 200 empty locations is populated by individual cells that exhibit a combination of the following characteristics: mating-type (0—empty, 1—P, 2—M), switching characteristic (2—non-switcher, 1—switcher that will switch, 0—switcher that will not switch; following the known switching pedigrees), ploidy (0—unmated haploid, 1—mated diploid). The number of locations occupied depends on a density variable. Each sexual cycle contains multiple rounds of asexual growth in which each cell can divide and occupy an empty neighbouring cell if available.

Switching follows the known S. pombe switching pedigree[24,74]. During each cell division, the inheritance of mating type depends on the switching status of the mother cell: the daughter cell remains of the same mating type if the mother cell is a non-switcher (0) or does not switch this round (2), and it will be of opposite mating type if the mother cell is switching (1). At random either the mother or the daughter cell migrates to the neighbouring cell, whereas the other cell remains in the previously occupied cell. During every asexual round, each cell has a certain chance R to grow (standard set to $R = 0.5$). A relative benefit to non-switching is introduced by scaling this chance by the selection coefficient s (with $0 > s > 0.5$) for non-switchers. Asexual growth cycles are iterated until every cell in the matrix is occupied.

To incorporate a cost for intraclonal mating, bookkeeping was performed to store if a cell was derived from a spore that arose from intraclonal mating or by outcrossing. A fitness cost for asexual growth was imposed by multiplying the previously estimated fitness with this cost.

When all cells are occupied, a randomly picked cell will be able to mate with a directly neighbouring cell (Moore neighbourhood), if there is a cell of opposite mating type present that has not yet mated in this round. Mating is continued until all cells that can mate have done so. Alternatively, mating is not restricted to the nearest neighbour but random within the whole grid.

After mating, all mated cells are counted and a new frequency of switchers and non-switchers is calculated from this. A new empty matrix is populated again with the same density according to the calculated switcher-non-switcher frequency from the previous round. Assuming Mendelian segregation at the mating-type locus, half of the cells are P and half are M. Switchers have equal chances of being in the switching or non-switching state. These cycles are repeated until either switcher or non-switcher become fixed, or after 500 transfers. The edges of the matrix are limited. We analysed the edge effect by implementing a torus shaped grid (connecting top with bottom and left with right), which did not show qualitative differences.

**Code availability.** The R script used for CA simulations is available from figshare at https://doi.org/10.6084/m9.figshare.5868528 (ref. [75]).

**Data availability.** Counts and frequencies from competitions are available from figshare at https://doi.org/10.6084/m9.figshare.5868528 (ref. [75]). Sequencing data

on evolved strain EBC7 are available at NCBI Sequence Read Archive, BioProject ID PRJNA438043; raw SMRT sequencing reads and de novo assembly under accession numbers SRP135217 and PYFU01000000, respectively.

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

## Acknowledgements

We thank Olaf Nielsen for providing the *h+S* strain, EG441 originally from Herbert Gutz strain collection and Ken Sawin for providing strain pKS394. Natural isolates were provided by Jürg Bähler. We also thank Bernadette Weissensteiner and Natalia Zajac for help with experimental work. This research was supported by grants from the Carl Tryggers Foundation, the Swedish Research Council, the European Research Council (HapSelA-336633) and by Ludwig-Maximilians University of Munich (LMU).

## Author contributions

B.N., S.T., J.W. and S.I. designed the study; B.N., S.T. and J.S. performed experiments; P. B. gave technical support and conceptual advice; BN designed and implemented the simulation model; ST performed the population genomics analyses; B.N. and S.T. performed the PacBio genomic analyses; B.N., S.T., J.W. and S.I. wrote the manuscript.

## Additional information

**Competing interests:** The authors declare no competing interests.

