## [Peer Review File · Nature Communications]

Reviewers' comments:

Reviewer #1 (Remarks to the Author):

This paper explores the evolution of mating type switching in the yeast *Saccharomyces pombe*, using both a phylogenetic and an experimental evolution approach. The phylogenetic approach shows that switching has been lost several times independently in natural populations. Using experimental evolution, the authors demonstrate that switching can evolve in an initially non-switching population in which both mating types are present, when individuals have to mate at regular intervals. Selection for switchers is stronger at lower population densities and shows positive frequency-dependence, which is interpreted as reflecting an advantage of intraclonal selfing for reproductive assurance (the authors also present results from a simulation model leading to the same conclusion). Finally, the authors demonstrate a cost of switching in asexually growing populations, and discuss several possible mechanisms for this cost.

The results are interesting and seem quite convincing (however I have no expertise in yeast biology nor in bioinformatics), and the paper is clearly written. My main comment is that I have found the Discussion a bit short on the biological implications of the results (concerning the evolution of mating type switching in natural populations). In particular, the results seem to imply that switching should be maintained in conditions where a substantial proportion of sexual spores are produced by intraclonal selfing (which should be effectively equivalent to asexual reproduction). Is there any evidence that intraclonal selfing is frequent in natural populations? Are there any strains able to generate dispersing spores asexually?

Furthermore, Figure 1 shows that switching has been lost several times in *S. pombe*, with little evidence for reversions. At the end of p.11, the authors argue that switching may be lost in conditions where populations reproduce asexually. If this is the case, it may be difficult to re-evolve switching (for example when one of the mating types has been lost): in a sense, the experiment shown here represents the best possible situation for the evolution of switching (starting from a population where both mating types are present in equal frequencies), but I guess that the evolution of switching is probably more difficult in nature.

Based on the results shown here, switching should be disfavoured in conditions where finding a compatible mate is not problematic. Are there any known populations of non-switching, sexually reproducing (and obligately outcrossing) yeasts? Also, could intraclonal mating be made possible by other mechanisms than mating type switching?

A more specific comment: the authors argue that switching is favoured because it allows reproductive assurance, but I have been wondering to what extent the automatic advantage of selfing, in terms of increased genetic transmission to the offspring (often discussed in the literature on the evolution of selfing) could also play a role.

Minor points:

5th line of the introduction: "reduce the probability of encountering a sexual partner by 50%": the reduction can be lower when more than 2 mating types are present.

line 334: "which does not allow the purging of deleterious mutations" seems a bit strong, perhaps replace by "which reduces the efficiency of selection against deleterious mutations"

Reviewer #2 (Remarks to the Author):

This paper investigates the evolution of mating type switching in fission yeast. There are three parts. The first documents on a phylogeny that mating type switching is a fast evolving trait. It gives genetic evidence for the occurrence of multiple events. The second part uses experimental evolution and fitness measures to show that switching is favored at low density and show positive frequency dependent selection. The last part (although presented only in 'discussion') includes numerical simulations to show how spatial patterning can explain the density effect. Some extra small (but nice) experiments are made to show spatial patterning in a real test case.

I find the paper overall convincing, and quite comprehensive. In an era where authors try to slice their findings in least publishable units, this is nice and refreshing. My deep congratulations for this. I have no major disagreement with the idea and evidence presented, so I think this study is very interesting, especially since this kind of question is not easily addressed. I strongly recommend publication, pending some revisions, which are all feasible.

I list below my concerns and several suggestions that can certainly be addressed in a revision. I have restrained myself, and tried to keep only the most important ones as the paper presents a lot of material.

Polymorphism

The paper insists on the occurrence of a polymorphism in switching and non-switching (see e.g. l86). This is not explained in the paper and discussed in a very confusing way. Showing the occurrence of cost and benefits does not lead to important polymorphism (see unclear statement l271-273). Mutation selection argument are provided l110-113), but they are not clear. Of course the frequency of a deleterious mutation can be larger at equilibrium than its rate of occurrence, provided it is not lethal. However, it is not clear if this is quantitatively adequate (see e.g. bold statement l116). Positive frequency dependence does not maintain polymorphism either. The simulations do not show polymorphism, confirming these points. This is repeated l360. Your simulations and results do not show maintenance of polymorphism. Why do you conclude the exact opposite? So what explains this polymorphism in natura? Please provide better and more consistent explanations.

Quantification

I find that the paper is weakest on issues related to quantification. Selection coefficient are not computed in a clear way. Standard population genetics measures based on frequency

change must be used to compute proper s (they are based on logistic frequency changes and are necessary to look for frequency dependent effects). Provide equations, definition and references for measuring selection coefficients. Do you have data for intermediate steps in the 'continuous exp' or 'repeated to stationary' protocols? How is frequency dependence corrected? Are initial frequencies well controlled?

Use frequencies rather ratios which have bad statistical properties. For instance, Fig 3 should show selection coefficient against frequency. Please also use the same focal type (switcher or non switcher) on both x and y axis (so that positive frequency dependence actually looks a positive trend). Explain better what are the "exceptions" indicated by numbers above boxes. This is too obscure. Comment on the outlier at 1:9 / 1:1000 on panel A. On panel B, explain better if there are other genetic differences among strains besides mCherry and mat locus. Where are the values larger than unity mentioned in the legend?

In the text, this lack of rigor should also be corrected, this is impeding proper understanding. For instance l236: ratio of what? l240-241 Sentence devoid of quantitative meaning. what is showing positive frequency dependence?

An important missing part in the simulation section is whether you reproduce the positive frequency dependence pattern and the observation that the strength of this dependence decreases with decreasing density (see nice pattern on Fig 3A). It would add significantly to the paper to consider this. How do the simulations results relates to the observed patterns (direction of frequency dependence, overall effect of density and effect of density on frequency dependence)?

Last point about quantification: you have the final frequency of switchers in all the replicate populations (including 11 pop with zero). You have selection coefficients and their dependence on density and frequency. You should be able to figure out what is approximately the mutation rate to a switching phenotype (the computation is not trivial, but feasible). This would quantify your argument about 'constraints'.

Sex versus switching

The paper conflates quite generally sex and switching. For instance l115, switching is said to be highly beneficial for surviving sexual reproduction. This is a confusing way to put it, even though I think I see what the authors want to say. It is clearer to say that it provides reproductive insurance during episodes of sex. If "survival to sex" was an issue, then, avoiding sex altogether would be enough!

Similarly, l329, in simulations, non switching would also invade if sex (or resistant spore stage) is not required. It is unclear quite generally why asexual resistant spores cannot be selected for. This issue is also touch upon when discussing the population genetics benefit of sex l333-l349. What did you do exactly when incorporating a cost for intraclonal selfing? Is this 'cost' carried over multiple generations? This is absolutely not equivalent to actually model the consequences of having deleterious mutations at many loci, which introduce intergenerational effects, and difference in both mean and variance in fitness in sex/asex progeny. Incorporating a real cost for intraclonal mating should make a difference, as intraclonal mating is useless in terms of sex. If there is no any evolutionary benefit of sex, then of course, doing intra or inter clonal mating is not different. Further, doing intraclonal mating can potentially reduce the scope for outcrossing, even at boundaries, where

alternative mates are available, provided outcrossing is not 'preferential'. These questions are confusing throughout. A last example is I347: strong directional selection has often been shown to favor sex, not asex.

Minor comments

abstract: the sentence "theoretical simulations elucidated ..." is, by quite far, an overstatement. The benefit and costs of sex were not really considered or modelled here. I47. 50% only true for two mating types. Also conflating sexes and mating types, which evolve probably for very different reasons (anisogamy, mt transmission vs inbreeding avoidance) is not necessarily illuminating.

I65. The qualitative difference is not clear if selfing and "universal compatibility" are both selected for reproductive insurance. Explain better.

I76. "selection dynamics" : very unclear meaning.

I77. again, "lonely spore" and "mating type ratio restoration" hypotheses seem both relying on the same reproductive insurance idea. Explain better

I82. "expected life cycle" is very unclear. The bottleneck argument is also unclear. Why isn't there also asexual bottlenecks?

I115. Sentence unclear.

I132. Explain better what are these deviations, and why they are interesting and why they differ from previous observations. This part is a bit superficial.

I201. Why would one expect some constraints?

I280. Unclear what is this repair. Explain better. The whole argument in the following paragraph on the origin of the cost is particularly unclear. Why should these costs occur if switching is cued on the same environmental trigger than sex?

I304-307. Argument about frequency dependence obscure. Explain better.

Suplem. Fig S4. you should use these results to say somewhere that results were not the results of contamination. The labelling of strains makes it suspicious: why most the evolved switchers have consecutive labelling (EBC24 to EBC30). what are the names of the strains which did not evolve switching?

Reviewer #3 (Remarks to the Author):

In this manuscript, Nieuwenhuis et al. perform an experimental evolution on the mating-type switching phenotype in fission yeast. They begin by showing that mating-type switching is prevalent in wild isolates of *S. pombe*, and their mechanism of switching is likely to be diverse. They then proceed to experimentally evolve switching phenotype from replicate non-switching lines and obtain reproducibly switching phenotypes after 25 transfers. To understand how switchers arise, they produce fitness measurements of switchers under different conditions, and finally produce a simulation that is used to explain the magnitude of the selection coefficients. Overall the study is exciting, and provides empirical evidence for the large diversity of mating type switching mechanism that is observed in different yeast species.

However, there are a few times where it is hard for the reader to understand the selection regime of the experiment and the validations performed to explain the observations. For

example, I list here a few times where it is hard to justify whether the validations performed for the fitness measurements are adequate for the experimental evolution.

Major clarifications needed:

1) The Figure 3 experiment doesn't even come close to testing the correct relevant range. The authors perform an experiment at a dilution of 1%, which is 1 million cells per petri dish (fig 2). The density, in cells/mm² is never indicated, but can be calculated ($\sim 1.50 \times 10^2$ cells per mm²). In Fig 3, the cell densities are written in cells/mm² ($\sim 8.0 \times 10^5$ cells per mm²). This is 5000 times more cells at the undiluted state than they did for their experiment (did the authors make a typo here? Did they really plate several mLs of cultures?). Their strongest dilution in fig 3 (1:1000) is not even close to the actual experiment (1:5000).

Nevertheless, this issue is not very pressing because the authors claim is likely that the phenotype would have also emerged at a 1:1000 dilution because stronger dilutions yield higher selection coefficients for the switching phenotype. More critical is that a mutation arising de novo in culture would be present at a ratio of 1 million:1 NS:SW. I am having a lot of trouble extrapolating from the Fig 3 what the selective effect of the switching phenotype would be at this ratio and at the correct density because the authors test a ratio that is orders of magnitude different (9:1). These distinctions are important when dealing with positive frequency dependent phenotypes. This is confounded by the second clarification needed:

2) The authors provide no estimate of the rate of mutation that yields the switching phenotype.

Under a very small rate of mutation to the switching phenotype, switchers may arise on the diploid plate in one cell and can be lost due to drift or selection during the asexual growth stage. Given a possible very small selection effect at this ratio (1:1 million), the phenotype is likely not to be expected to fix under their selection regime. At an $s \sim 80\%$, the phenotype is expected to fix in about 25 generations if the mutation arises at the first generation and is not lost to drift, but if the selective effect is much smaller at earlier generations it becomes unclear if fixation is expected. I believe this implies that the rate of mutation to this phenotype must be very high. We might therefore expect several different mechanisms within a single population with the phenotype, which is corroborated with the multiple mechanisms across populations.

Could the authors provide a simple estimate of the mutation rate to switching? Perhaps plating high densities of both mating types on a plate, then spreading out the spores to single colonies and having an assay that can quickly verify if a colony is a switcher? I am not familiar with working in *S. pombe* but I think providing an estimate of the mutation in absence of selection is likely to be important.

3) I think the simulation is interesting, and has power to address some of the concerns I listed previously. However, I cannot understand if the simulation is within the relevant regime. Is 40% of the area as starting density the same as their experiment?

4) The frequency dependence in asexual growth experiment is also necessary for the switcher vs non-switcher test in order to have an intuition on the actual cost of switching during the actual experimental evolution. I understand that frequency dependence here is perhaps not expected due to equal mating type ratios, but can be tested to confidently assert this.

Minor concerns:

- 1) There seems to be a typo in the selection coefficients of switching under asexual growth: line 265 indicates a cost of 0.5%, and line 342 indicates a cost of 0.05%.
- 2) How is this normalized relative change calculated? Can the authors provide the formula? I suspect that it is the logn of the ratio of the logits since that is usually the selection coefficients but it is not clear here. Can the authors also provide the formula for how they corrected the marker effect? Figure S7/S8 have a typo on the y-axis (panel b says marker-corrected but it is marker uncorrected).

The experimental design for the experimental evolution is quite nice, and the few cases where they PCR the genotypes to show multiple mechanisms of switching is great. However, it's not abundantly clear to me whether the switchers could be fixed in their experiment if the mutation rate to the switching phenotype is rare due to the frequency dependence on the selective effect of the switching phenotype. If the mutation to switching is very common, then perhaps their story about how switching/non-switching evolved is less compelling. I believe all these issues could be easily fixed by reproducing their experiments under a more adequate regime of a new mutation entering the population in a ratio much much lower than 9:1, and providing a fluctuation-type assay on the accumulation of this switching phenotype.

Reviewer #1 (Remarks to the Author):

1. This paper explores the evolution of mating type switching in the yeast *Saccharomyces pombe*, using both a phylogenetic and an experimental evolution approach. The phylogenetic approach shows that switching has been lost several times independently in natural populations. Using experimental evolution, the authors demonstrate that switching can evolve in an initially non-switching population in which both mating types are present, when individuals have to mate at regular intervals. Selection for switchers is stronger at lower population densities and shows positive frequency-dependence, which is interpreted as reflecting an advantage of intraclonal selfing for reproductive assurance (the authors also present results from a simulation model leading to the same conclusion). Finally, the authors demonstrate a cost of switching in asexually growing populations, and discuss several possible mechanisms for this cost.

The results are interesting and seem quite convincing (however I have no expertise in yeast biology nor in bioinformatics), and the paper is clearly written. My main comment is that I have found the Discussion a bit short on the biological implications of the results (concerning the evolution of mating type switching in natural populations). In particular, the results seem to imply that switching should be maintained in conditions where a substantial proportion of sexual spores are produced by intraclonal selfing (which should be effectively equivalent to asexual reproduction). Is there any evidence that intraclonal selfing is frequent in natural populations? Are there any strains able to generate dispersing spores asexually?

We thank the reviewer for their valuable comments and positive review of our manuscript. We have expanded the discussion to further highlight the biological implications of our findings, specifically addressing i) the expected reproduction rates occurring asexually by mitotic growth, or sexually by ii) selfing and iii) outcrossing.

i) Mutants that can sporulate asexually are known, but they are almost all aneuploid and have strongly reduced viability (Iino & Yamamoto 1985 MGG doi: 10.1007/BF00332932.). No natural isolates are known to exhibit asexual sporulation. Production of resistant cells is in most fungi completely linked to the sexual cycle, which we address in the first paragraph of the discussion (lines 268) *“In most fungi, production of resistant cells is completely linked to the sexual cycle and thus fully depends on successful mating.”* and when discussing the consequences of homothallism *“The fact that no yeasts have evolved euploid survival propagules without recombination suggests that sexual reproduction is an indispensable component of spore formation”* (lines 327)

ii) As pointed out by the reviewer, it is difficult to distinguish intraclonal mating from asexual reproduction. For budding yeast, this has been measured. One might expect that due to their similarity in lifecycle and external factors that induce sporulation, the frequency of meiotic reproduction is similar for fission and budding yeast. In budding yeast, meiosis is estimated to occur about once per 1000 mitotic generations and outcrossing once per

~100 000 generations (Tsai et al 2008 PNAS, doi: 10.1073/pnas.0707314105). Implementing our findings on the cost of switching during asexual reproduction and the observed occurrence of switchers, we argue that mating in fission yeast should be more common. We added this information to the Discussion when describing the role of outcrossing and switching: *“The observed frequency of 15% non-switchers suggests that sexual reproduction, which strongly selects against loss of switching, should occur more regularly to avoid loss of switchers.”* (lines 330-340).

iii) Linkage disequilibrium in fission yeast is much higher than in budding yeast, which suggests a lower outcrossing rate (~1 per 600 000 generations; Farlow et al 2015 Genetics, doi: 10.1534/genetics.115.177329). We added this information to the discussion *“outcrossing rates in fission yeast in nature, based on linkage disequilibrium, is estimated to occur only once per 600 000 generations”* (lines 320-323).

2. Furthermore, Figure 1 shows that switching has been lost several times in *S. pombe*, with little evidence for reversions. At the end of p.11, the authors argue that switching may be lost in conditions where populations reproduce asexually. If this is the case, it may be difficult to re-evolve switching (for example when of the mating types has been lost): in a sense, the experiment shown here represents the best possible situation for the evolution of switching (starting from a population where both mating types are present in equal frequencies), but I guess that the evolution of switching is probably more difficult in nature.

This is an interesting point. It is true that in our experiment both mating types are continuously present. Hence, its reversion is likely more efficiently obtained than if only one mating type would prevail. This may not always be the case in nature but in *S. pombe*, reversion to switching might occur readily, because the most common non-switching genotype (h+N) maintains genetic information of both mating types and naturally reverts to a switching genotype roughly once every 10 000 cell divisions. We address the nature of this structural mutation and its rate more extensively in lines 349-350. *“This is even possible when locally only one mating type occurs, because in S. pombe a variety of heterothallic genotypes spontaneously revert to switching.”*

3. Based on the results shown here, switching should be disfavoured in conditions where finding a compatible mate is not problematic. Are there any known populations of non-switching, sexually reproducing (and obligately outcrossing) yeasts? Also, could intraclonal mating be made possible by other mechanisms than mating type switching?

While in theory we may expect heterothallic yeasts to exist at high enough population densities, most yeasts are either mating-type switching (e.g. Saccharomycetes and Schizosaccharomycetes, doi: 10.1534/genetics.117.202036), homothallic (e.g.

Pneumocystidomycetes and *Cochliobolus* spp. doi: 10.1128/mBio.00583-15) or heterothallic with diploid/dikaryotic dispersal (many Basidiomycetes doi:10.1128/EC.00440-07, *Kazachstania* spp. doi: 10.1111/j.1567-1364.2006.00163.x). Heterothallic yeasts are described in the literature, but often have alternative lifecycles in which zygotes produce diploid spores (e.g. *Kazachstania* spp. *ibid*) or where dispersal is later shown to occur both as diploid as well as haploid spores, possibly as a bet hedging strategy (e.g. *Starmera* spp. doi: 10.1099/ijsem.0.000596).

Other intra-clonal mating mechanisms evolved, probably more readily than mating type switching. We now emphasize this aspect when introducing homothallism (lines 61-71).

4. A more specific comment: the authors argue that switching is favoured because it allows reproductive assurance, but I have been wondering to what extent the automatic advantage of selfing, in terms of increased genetic transmission to the offspring (often discussed in the literature on the evolution of selfing) could also play a role.

The automatic advantage hypothesis of selfing is assumed an important driver of the evolution of selfing in organisms where one of the two sex roles are limiting. For example in flowering plants, pollen allows for outcrossing while ovules can be self-fertilized, a process which allows insurance of reproduction without reducing the total number of seeds produced. This solution is also possible in haploid and diploid fungi where both male and female sex roles exist during mating (i.e. these are hermaphroditic), but not in isogamic haploid yeasts where each mating is symmetrical. We refer shortly to this in lines 67-71. *"The automatic advantage of selfing under which all female gametes are self-fertilized while additional fertilization by outcrossing through male gametes continues, might apply to anisogametic, multicellular fungi. In single celled haploid organisms with isogametic gametes such as yeasts, a contribution of the automatic advantage for the evolution of homothallism can be excluded."*

Minor points:

5. 5th line of the introduction: "reduce the probability of encountering a sexual partner by 50%": the reduction can be lower when more than 2 mating types are present.

Good point, this has been corrected (Line 47).

6. line 334: "which does not allow the purging of deleterious mutations" seems a bit strong, perhaps replace by "which reduces the efficiency of selection against deleterious mutations"

Changed as suggested.

Reviewer #2 (Remarks to the Author):

7. This paper investigates the evolution of mating type switching in fission yeast. There are three parts. The first documents on a phylogeny that mating type switching is a fast evolving trait. It gives genetic evidence for the occurrence of multiple events. The second part uses experimental evolution and fitness measures to show that switching is favored at low density and show positive frequency dependent selection. The last part (although presented only in ‘discussion’) includes numerical simulations to show how spatial patterning can explain the density effect. Some extra small (but nice) experiments are made to show spatial patterning in a real test case. I find the paper overall convincing, and quite comprehensive. In an era where authors try to slice their findings in least publishable units, this is nice and refreshing. My deep congratulations for this. I have no major disagreement with the idea and evidence presented, so I think this study is very interesting, especially since this kind of question is not easily addressed. I strongly recommend publication, pending some revisions, which are all feasible. I list below my concerns and several suggestions that can certainly be addressed in a revision. I have restrained myself, and tried to keep only the most important ones as the paper presents a lot of material.

We thank the reviewer for his/her very positive evaluation of our manuscript.

Polymorphism

8. The paper insists on the occurrence of a polymorphism in switching and non-switching (see e.g. L86). This is not explained in the paper and discussed in a very confusing way. Showing the occurrence of cost and benefits does not lead to important polymorphism (see unclear statement L271-273). Mutation selection argument are provided (L110-113), but they are not clear. Of course the frequency of a deleterious mutation can be larger at equilibrium than its rate of occurrence, provided it is not lethal. However, it is not clear if this is quantitatively adequate (see e.g. bold statement L116). Positive frequency dependence does not maintain polymorphism either. The simulations do not show polymorphism, confirming these points. This is repeated L360. Your simulations and results do not show maintenance of polymorphism. Why do you conclude the exact opposite? So what explains this polymorphism in natura? Please provide better and more consistent explanations.

Thank you for pointing this out to us. The use of the word “polymorphism” is not ideal as it might suggest maintenance of both strategies in the same population. We do not think this is the case, but more likely that under alternating ecological circumstances either one or the other strategies will be fixed in a population, hence leading to the coexistence of the two strategies on the level of the species. We have removed the word polymorphism and extensively discuss the existence of both mating systems in the context of alternatively selected strategies in the fourth paragraph of the discussion (Lines 330-344).

Quantification

9. I find that the paper is weakest on issues related to quantification. Selection coefficients are not computed in a clear way. Standard population genetics measures based on frequency change must be used to compute proper s (they are based on logistic frequency changes and are necessary to look for frequency dependent effects). Provide equations, definition and references for measuring selection coefficients. Do you have data for intermediate steps in the ‘continuous exp’ or ‘repeated to stationary’ protocols? How is frequency dependence corrected? Are initial frequencies well controlled?

Use frequencies rather ratios which have bad statistical properties. For instance, Fig 3 should show selection coefficient against frequency. Please also use the same focal type (switcher or non-switcher) on both x and y axis (so that positive frequency dependence actually looks a positive trend).

We now explicitly calculate selection coefficients in all instances where fitness of switchers and non-switchers is compared. The competitions are analyzed using standard exponential growth calculations to estimate selection coefficients (following Lenski et al 1991) per mitotic fission event (lines 529-540) or per sexual generation (lines 573-579) for growth and mating competitions respectively. These analyses are described in the methods section. Normalization is no longer required.

We did not correct for frequency dependence. The change in frequency between the strains was small and we always started with equal ratios of competing types. We assume no large effect of frequency dependence as 1) the effect measured in Fig S10 was with much stronger initial skews and 2) no difference is observed when comparing competitions between a switching strain with both mating types or with only one mating type (fig 3 and S9).

10. Explain better what are the “exceptions” indicated by numbers above boxes. This is too obscure. Comment on the outlier at 1:9 / 1:1000 on panel A. On panel B, explain better if there are other genetic differences among strains besides mCherry and mat locus. Where are the values larger than unity mentioned in the legend?

The reduced number of replicates was explained in the methods section, we now directly refer to these in the figure caption. Due to the low cell numbers at the low density regime it becomes difficult to distinguish cells from background noise (cell debris and enzyme mix that arise when all asexual cells are destroyed) and only samples were used for which at least 1800 true counts after filtering were available (median counts per sample after filtering are 18000). This arbitrary cutoff resulted in the outlier. We improved the filtering method and were able to reduce the noise. The specific data point drops out with the improved filtering. The same conclusions are obtained with both filtering methods.

11. In the text, this lack of rigor should also be corrected, this is impeding proper understanding. For instance L236: ratio of what? L240-241 Sentence devoid of quantitative meaning. What is showing positive frequency dependence?

We carefully revised the entire text and added or clarified these and similar unspecific or vague statements throughout.

12. An important missing part in the simulation section is whether you reproduce the positive frequency dependence pattern and the observation that the strength of this dependence decreases with decreasing density (see nice pattern on Fig 3A). It would add significantly to the paper to consider this. How do the simulations results relates to the observed patterns (direction of frequency dependence, overall effect of density and effect of density on frequency dependence)?

Density dependence is seen in our simulations. Higher densities increase mating efficiency for non-switcher (Fig S11) and result in longer waiting times to fixation of the mating type switcher or under higher densities and with cost to switching, reduction in fixation time of the non-switcher. We added a supplementary figure S12 to show the change in time to fixation with respect to density dependence.

In terms of frequency-dependence we hypothesize that the result of frequency dependence found in our mating competitions is the result of preference by the mating type switchers to mate with a direct sister cell due to synchronization. We regard this aspect to be beyond the scope of the simulations, but discuss it further in lines 235-239.

13. Last point about quantification: you have the final frequency of switchers in all the replicate populations (including 11 pop with zero). You have selection coefficients and their dependence on density and frequency. You should be able to figure out what is approximately the mutation rate to a switching phenotype (the computation is not trivial, but feasible). This would quantify your argument about ‘constraints’.

This is a good suggestion, however, as the computation indeed is not trivial, we decided to measure an upper and calculate a lower estimate for the mutation rate. We empirically quantified the upper boundary of mutation rate ($10e-06$) and estimated a mutation rate under the assumption that under a high effective population size the effect of drift is negligible and that a highly beneficial mutant ($s=0.74$) will have a fixation probability of 1.

We performed the estimate for mutation of mating-type switching as suggested by reviewer 3 (see comment #35) by plating out 1 million spores derived from a cross between EBC1 and EBC3 (one of the two non-switching pairs of strains used in the experimental evolution) onto mating plates and screened these for spontaneous switcher mutations. Not a single switcher was observed. This suggests that the mutation rate is below 1 in approximately a million meioses.

We also verified the occurrence of multiple mating-type switchers evolving within a single population. We randomly picked 50 single spore colonies from the population that contained the PacBio sequenced strain and performed PCR to verify presence of the inversion of the silent *matP*, which occurred uniquely in this population. If another genotype would have evolved in this population it would be likely to be by a duplication as was observed in most other populations. All 50 strains showed the presence of the *matP* in the inverted direction, suggesting that in this population only one event led to the evolution of switching.

Additionally, we calculated a maximum mutation rate of $7.2 * 10^{-9}$ based on the results from the evolution experiment: 9 mutations got selected in 25 generations in 20 populations of size $1.25 * 10^6$ zygotes.

Overall, the relatively low mutation rate and the very high selection coefficient seem to fit the observation that not all populations evolved switching, and that in those that did, the frequency of switchers can increase rapidly and go to fixation within the 25 generations of our experiment.

The here described extra experiments and the calculation of the minimum mutation rate have been added to the methods section and the results are described in the Results section (lines 481-494 and 173-197 respectively).

Sex versus switching

14. The paper conflates quite generally sex and switching. For instance L115, switching is said to be highly beneficial for surviving sexual reproduction. This is a confusing way to put it, even though I think I see what the authors want to say. It is clearer to say that it provides reproductive insurance during episodes of sex. If “survival to sex” was an issue, then, avoiding sex altogether would be enough!

This is an important point, and we carefully revised the entire manuscript to clarify this aspect. We now clearly explain the importance of “sex” for survival (lines 56-57 and 266-270) and the importance of “switching” to be able to reproduce sexually (lines 334-344) and we use the terms consistently throughout.

15. Similarly, L329, in simulations, non-switching would also invade if sex (or resistant spore stage) is not required. It is unclear quite generally why asexual resistant spores cannot be selected for. This issue is also touched upon when discussing the population genetics benefit of sex L333-1349.

In many fungi (as well as in other facultative asexuals), the evolution of resistance is almost always linked to sexual reproduction (Hartfield, 2016 doi:10.1111/jeb.12770). It is

remarkable that no resistant asexual cells have evolved in yeasts, but apparently the link to meiosis is not easily broken, and/or under certain circumstances not as costly (Aanen & Hoekstra 2007 doi: 10.1128/9781555815837.ch32). Mutants that can sporulate asexually are known, but they are almost all aneuploid and have strongly reduced viability (Iino & Yamamoto 1985 MGG doi: 10.1007/BF00332932.). No natural isolates are known to exhibit asexual sporulation. We discuss this in the first paragraph of the discussion (lines 268) *“In most fungi, production of resistant cells is completely linked to the sexual cycle and thus fully depends on successful mating.”* and when discussing the consequences of homothallism *“The fact that no yeasts have evolved euploid survival propagules without recombination suggests that sexual reproduction is an indispensable component of spore formation”* (lines 327)

16: What did you do exactly when incorporating a cost for intraclonal selfing? Is this ‘cost’ carried over multiple generations? This is absolutely not equivalent to actually model the consequences of having deleterious mutations at many loci, which introduce intergenerational effects, and difference in both mean and variance in fitness in sex/asex progeny. Incorporating a real cost for intraclonal mating should make a difference, as intraclonal mating is useless in terms of sex. If there is no any evolutionary benefit of sex, then of course, doing intra or inter clonal mating is not different. Further, doing intraclonal mating can potentially reduce the scope for outcrossing, even at boundaries, where alternative mates are available, provided outcrossing is not ‘preferential’. These questions are confusing throughout.

This is an interesting point. The cost of intra-clonal selfing was modelled to last only for a single generation (described in the methods section, lines 600-603). The inoculum in the simulations that were used to inoculate a new round of mating were either the result of a selfing or of an outcrossing event, and in the former situation, a penalty during asexual growth was introduced for these strains. We did not extend on this, as the model did not easily allow for adding multi-generation bookkeeping.

If haploid-selfing is costly, it might be beneficial to evolve a non-self preference, however, this was not observed in our measurements of mating, which on the contrary suggested higher selfing than outcrossing. Additionally, in nature outcrossing is expected to occur very rarely; only once every 600 000 generations (Farlow et al 2015, doi: 10.1534/genetics.115.177329). We explain these interactions better in the Discussion (lines 317-329).

17. A last example is L347: strong directional selection has often been shown to favor sex, not asex.

Here we are referring to strong short-term selection when sexually produced resting spores are not desired, but where high growth rate is demanded. When no sex is required, these relatively expensive pathways are often sacrificed (e.g. Lang et al. 2011, doi: 10.1534/genetics.111.128942). We have modified the sentence to make this point clear (lines 354-358, *“Similarly, costs during asexual growth might also explain the strongly reduced frequency of mating type switching (e.g. C. glabrata) or even full loss of switching in opportunistic pathogenic fungi (e.g. Lachancea kluyveri) that mainly grow asexually and where the immune system is likely to impose strong short-term selection on locally adapted strains.”*)

Minor comments

18. abstract: the sentence “theoretical simulations elucidated ...” is, by quite far, an overstatement. The benefit and costs of sex were not really considered or modelled here.

We fully agree and modified the sentence to: *“Detailed fitness measurements accompanied by computer simulations show the benefits and costs of switching during sexual and asexual reproduction governing the occurrence of both strategies in nature.”* (lines 36-38).

19. L47. 50% only true for two mating types. Also conflating sexes and mating types, which evolve probably for very different reasons (anisogamy, mt transmission vs inbreeding avoidance) is not necessarily illuminating.

The reviewer makes a good point that mating types and sexes evolved for different purposes. However, the consequences of different sexes or types are similar in that they both reduce chances of finding a suitable mating partner. We made this explicit in the text and made the distinction between sexes and mating types (lines 45-47, *“Sexual systems requiring such mate compatibility reduce the probability of encountering a suitable mating partner by up to 50% when two sexes or two mating types exist.”*).

20. L65. The qualitative difference is not clear if selfing and “universal compatibility” are both selected for reproductive insurance. Explain better.

We have rewritten this statement (lines 65-67, *“Note that additional to reproductive assurance, homothallism might have evolved for the reduction of outbreeding depression, for increased outcrossing compatibility to generate variation, or for all three”*).

21. L76. “selection dynamics” : very unclear meaning.

Sentence is rephrased to *“[...], little is known about its evolutionary origin and the forces that drive selection of self-compatibility at the population level”* (lines 81-82)

22. L77. again, “lonely spore” and “mating type ratio restoration” hypotheses seem both relying on the same reproductive insurance idea. Explain better

We followed the suggestions of the reviewer and added “*both suggesting such an evolutionary response to be an adaptation for reproductive assurance*” (lines 84).

23. L82. “expected life cycle” is very unclear. The bottleneck argument is also unclear. Why isn’t there also asexual bottlenecks?

We rewrote this section to explain more clearly the lifecycle and when bottlenecks are expected (lines 85-93).

24. L115. Sentence unclear.

Rephrased to “*Here we investigated the molecular, evolutionary and ecological underpinnings for the emergence and persistence of switching versus non-switching strategies [...]*” (lines 101-102).

25. L132. Explain better what are these deviations, and why they are interesting and why they differ from previous observations. This part is a bit superficial.

We rephrased the sentence. The deviations were merely strains that were wrongly assigned switching or non-switching in a previous study. We did not do further analyses on these strains. “*We analysed the strains described in Jeffares et al. by iodine staining, PCR amplification of the mat1 locus, and test crosses, to obtain information for all 57 strains and found some inconsistencies with previously described observations (see the first and second column in Fig. 1b, Supplementary Table S13).*” (lines 120-124)

26. L201. Why would one expect some constraints?

We rephrased the sentence.

27. L280. Unclear what is this repair. Explain better. The whole argument in the following paragraph on the origin of the cost is particularly unclear.

27. We extended the explanation of the potential costs of mating type switching, explaining the DNA damage induced during mitosis, as well as more extensively explaining the effect of pheromones and the cost of inheritance of the ‘wrong’ mating -type specific proteins (lines 275-298).

28. Why should these costs occur if switching is cued on the same environmental trigger than sex?

This is a good point. It seems that this would be an optimal strategy and indeed, some methylotrophic yeast species do exactly that. These go through one mitotic division in which

switching occurs, directly followed by mating (Hanson et al 2014, doi:10.1073/pnas.1416014111). The regulatory pathway for meiosis in these species was largely rewired compared to the other budding yeasts, which might have made this link possible. In *S. pombe*, due to some constraints in the mating pathway, this might have hampered its evolution. We refer to this strategy in the discussion (lines 358-359). *“An alternative to reducing costs is to switch only when mating is required, as occurs in some methylotrophic yeast where switching is induced by the same cue as mating”*

29. L304-307. Argument about frequency dependence obscure. Explain better.

We extended the explanation of the effect of mating-type pheromones on the growth and how a skew in mating-type ratios changes the presence of one of the mating type pheromones that in effect reduce growth of the minority mating type. (lines 281-290).

30. Suplem. Fig S4. you should use these results to say somewhere that results were not the results of contamination. The labelling of strains makes it suspicious: why most the evolved switchers have consecutive labelling (EBC24 to EBC30). what are the names of the strains which did not evolve switching?

The labelling of strains is purely chronological and not restricted to the here described project. Strain EBC7 was isolated first to be sequenced using PacBio. The other strains were isolated few months later (hence the gap in numbers). The population from which strain EBC141 was isolated was initially assumed to be completely non-switching due to the low frequency of switchers and the few colonies tested. In a later more thorough screen to measure frequencies (used to generate Suppl Fig S4a), switchers were observed and a clone was isolated. We added information about this non-consecutive numbering to Supplementary Table S2.

Reviewer #3 (Remarks to the Author):

31. In this manuscript, Nieuwenhuis et al. perform an experimental evolution on the mating-type switching phenotype in fission yeast. They begin by showing that mating-type switching is prevalent in wild isolates of *S. pombe*, and their mechanism of switching is likely to be diverse. They then proceed to experimentally evolve switching phenotype from replicate non-switching lines and obtain reproducibly switching phenotypes after 25 transfers. To understand how switchers arise, they produce fitness measurements of switchers under different conditions, and finally produce a simulation that is used to explain the magnitude of the selection coefficients. Overall the study is exciting, and provides empirical evidence for the large diversity of mating type switching mechanism that is observed in different yeast species.

We thank the reviewer for this positive assessment of our study.

32. However, there are a few times where it is hard for the reader to understand the selection regime of the experiment and the validations performed to explain the observations. For example, I list here a few times where it is hard to justify whether the validations performed for the fitness measurements are adequate for the experimental evolution.

We thank the reviewer for their comments. We carefully revised our manuscript to address the issues raised by the reviewer and performed two additional analyses to further explain our experimental evolution results.

Major clarifications needed:

33. 1) The Figure 3 experiment doesn't even come close to testing the correct relevant range. The authors perform an experiment at a dilution of 1%, which is 1 million cells per petri dish (fig 2). The density, in cells/mm² is never indicated, but can be calculated ($\sim 1.50 \times 10^2$ cells per mm²). In Fig 3, the cell densities are written in cells/mm² ($\sim 8.0 \times 10^5$ cells per mm²). This is 5000 times more cells at the undiluted state than they did for their experiment (did the authors make a typo here? Did they really plate several mLs of cultures?). Their strongest dilution in fig 3 (1:1000) is not even close to the actual experiment (1:5000).

The reviewer is correct that the number of cells in the highest concentration used to measure the effect of cell density on the mating success of switchers vs. non-switchers is far higher than in the evolution experiment. However, the purpose of these measurements was to analyze the importance of density on mating, not to just repeat part of the evolution experiment. Though the density in the evolution experiment was intermediate to the lowest two densities.

The highest concentration used for Fig 3 was derived from milliliters of cell culture, that were concentrated by spinning down the cells which (i.e. not a typo). This concentration was chosen to assure all cells were in direct contact with multiple other cells. The successive dilutions indicate an increased fitness benefit with reduced density.

The density we used in the experiment ($\sim 3.1 \times 10^3$ cells/mm²; we used a smaller plate than the reviewer assumed and plated out a larger number of cells; this is described in the methods section) was intermediate to the two highest dilutions. The values obtained in these competitions match well with the findings from the evolution experiment. These details and the calculations are described in the Methods section (lines 438 and 559).

34. Nevertheless, this issue is not very pressing because the authors claim is likely that the phenotype would have also emerged at a 1:1000 dilution because stronger dilutions yield higher selection coefficients for the switching phenotype. More critical is that a mutation arising de novo in culture would be present at a ratio of 1 million:1 NS:SW. I am having a lot of trouble extrapolating from the Fig 3 what the selective effect of the switching phenotype would be at this ratio and at the correct density because the authors test a ratio that is orders of magnitude

different (9:1). These distinctions are important when dealing with positive frequency dependent phenotypes.

The referee makes a good point that positive frequency dependence might hamper invasion of a novel mutation. However, we find positive frequency dependence for the switchers, but this frequency dependence is only relevant at very high cell densities. Mating type switching at lower dilutions is always beneficial, irrespective of the initial frequency. What our results show is that non-switching during sexual reproduction can only be maintained under the highest densities and then only when there are a lot of non-switchers present to mate with. At low switcher frequencies the total number of non-switchers that can mate is low, but a switcher will always be able to reproduce. The absolute fitness of the switcher is thus not frequency dependent and only to a lesser degree density dependent, but that of the non-switcher is. The apparent frequency dependence in *relative* fitness will thus mostly be affected by the non-switchers. We now explain the role of frequency dependence in more detail in the results section (lines 230-239).

35. This is confounded by the second clarification needed:

2) The authors provide no estimate of the rate of mutation that yields the switching phenotype. Under a very small rate of mutation to the switching phenotype, switchers may arise on the diploid plate in one cell and can be lost due to drift or selection during the asexual growth stage. Given a possible very small selection effect at this ratio (1:1 million), the phenotype is likely not to be expected to fix under their selection regime. At an $s \sim 80\%$, the phenotype is expected to fix in about 25 generations if the mutation arises at the first generation and is not lost to drift, but if the selective effect is much smaller at earlier generations it becomes unclear if fixation is expected. I believe this implies that the rate of mutation to this phenotype must be very high. We might therefore expect several different mechanisms within a single population with the phenotype, which is corroborated with the multiple mechanisms across populations. Could the authors provide a simple estimate of the mutation rate to switching? Perhaps plating high densities of both mating types on a plate, then spreading out the spores to single colonies and having an assay that can quickly verify if a colony is a switcher? I am not familiar with working in *S. pombe* but I think providing an estimate of the mutation in absence of selection is likely to be important.

We thank the reviewer for this excellent suggestion. We now added calculations for the selection coefficient for switchers which showed a selection coefficient of 0.74 during sexual reproduction at the lowest dilution, which we assume is an underestimate due to the high noise levels at this dilution (see reply to comment #10)

The fixation probability of such a highly beneficial mutation will thus be close to unity too. An s of 0.74 (probably an underestimate) suggests that fixation of a beneficial mutation can

occur within 19 generations in our population with an effective population size of $1/4 * 5 * 10^6$ (total number of meioses = 1/4 of the total number of spores produced).

We performed the estimate for mutation of mating-type switching as suggested by the reviewer, and plated out 1 million spores derived from a cross between EBC1 and EBC3 (one of the two non-switching pairs of strains used in the experimental evolution) onto mating plates and screened these for spontaneous switcher mutations. Not a single switcher was observed. This suggests that the mutation rate is below 1 in approximately a million meioses.

We also verified the occurrence of multiple mating-type switchers evolving within a single population. We randomly picked 50 single spore colonies from the population that contained the PacBio sequenced strain and performed PCR to verify presence of the inversion of the silent *matP*, which occurred uniquely in this population. If another genotype would have evolved in this population it would be likely to be by a duplication as was observed in most other populations. All 50 strains showed the presence of the *matP* in the inverted direction, suggesting that in this population only one event led to the evolution of switching.

Additionally, we calculated a maximum mutation rate of $7.2 * 10^{-9}$ based on the results from the evolution experiment: 9 mutations got selected in 25 generations in 20 populations of size $1.25 * 10^6$ zygotes (see also comment #13 from reviewer 2).

Overall, the low mutation rate and the very high selection coefficient seem to fit the observation that not all populations evolved switching, and that in those that did, the frequency of switchers can increase rapidly and go to fixation within the 25 generations of our experiment.

The here described extra experiments and the calculation of the minimum mutation rate have been added to the methods section and the results are described in the Results section (lines 481-494 and 173-197 respectively).

36. 3) I think the simulation is interesting, and has power to address some of the concerns I listed previously. However, I cannot understand if the simulation is within the relevant regime. Is 40% of the area as starting density the same as their experiment?

The values of our evolution experiment are at the low end of density used in the simulations (left side of the figure). The simulations show that in our evolution experiment whenever switching evolves it should go to fixation, as only under very high densities and when

switching is very costly will non-switching be maintained. We have improved the integration of our simulations and the experiment in our Discussion section. (lines 307-329)

37. 4) The frequency dependence in asexual growth experiment is also necessary for the switcher vs non-switcher test in order to have an intuition on the actual cost of switching during the actual experimental evolution. I understand that frequency dependence here is perhaps not expected due to equal mating type ratios, but can be tested to confidently assert this.

As rightly pointed out by the reviewer, in our experiment, the mating type ratios are equal, and thus the frequency dependence is of little effect during the growth phase of our evolution experiment. On the mating plate however, a difference in growth might occur, but only when growth is very costly for switchers will this affect the outcome of the competitions, as can be seen from the simulations. The growth rate differences between switchers and non-switchers we measured were far from levels required for fixation of the non-switcher and also the densities in the experiment were lower. We therefore decided to not further delve in to this effect.

Minor concerns:

38. 1) There seems to be a typo in the selection coefficients of switching under asexual growth: line 265 indicates a cost of 0.5%, and line 342 indicates a cost of 0.05%.

Thank you for spotting this mistake. As mentioned above (see also our answer to comment #39), we have recalculated our results to selection coefficients and these values are no longer used.

39. 2) How is this normalized relative change calculated? Can the authors provide the formula? I suspect that it is the logn of the ratio of the logits since that is usually the selection coefficients but it is not clear here. Can the authors also provide the formula for how they corrected the marker effect?

We have recalculated all selection coefficients as described by the reviewer. These calculations are described in the methods sections (lines 529-540 for asexual and 573-579 for sexual competitions). To compensate for the marker effect, we calculated the selection coefficient based on the effect caused by the marker and the effect caused by switching to be additive ($S_{\text{switching}} = S_{\text{total}} - S_{\text{marker}}$). The effect of the marker was measured using competitions between isogenic strains that differed in the marker under equal circumstances and densities as the competitions. Tests in which competitions were performed in both directions (switcher marked competing with non-switcher wildtype and vice-versa) confirm this (figures S9 and S10).

40. Figure S7/S8 have a typo on the y-axis (panel b says marker-corrected but it is marker uncorrected).

Thank you for spotting this typo – we corrected it accordingly.

41. The experimental design for the experimental evolution is quite nice, and the few cases where they PCR the genotypes to show multiple mechanisms of switching is great. However, it's not abundantly clear to me whether the switchers could be fixed in their experiment if the mutation rate to the switching phenotype is rare due to the frequency dependence on the selective effect of the switching phenotype. If the mutation to switching is very common, then perhaps their story about how switching/non-switching evolved is less compelling. I believe all these issues could be easily fixed by reproducing their experiments under a more adequate regime of a new mutation entering the population in a ratio much much lower than 9:1, and providing a fluctuation-type assay on the accumulation of this switching phenotype.

Please see our answer (#35) above about the effect of mutation rates of novel mating type switching genotypes and the fixation rates of these highly selected traits. Our competition experiments show that under lower densities, switchers have an increasing benefit over non-switchers. Frequency dependence mainly plays a role at high densities, because it increases the mating potential of the non-switchers (there will be more borders between patches of NS of opposite mating types) and thus the ratio of mated non-switcher borders vs mated switcher patches will increase. Under low densities, this effect becomes minimal, because there are very few patches and hence very few borders. The size of each patch is independent of frequency, only the number of borders between non-switchers changes from moderate to very low.

Reviewers' comments:

Reviewer #1 (Remarks to the Author):

My comments have been satisfactorily addressed by the authors; I don't have any remaining criticism and recommend acceptance.

Reviewer #2 (Remarks to the Author):

The revision is very well done. the text is clearer and to the point. The only place where the revision is still a bit unclear is related to the computation of selection coefficients. Otherwise, I'm happy with the revision. Congratulation for this!

The equations used are somewhat complicated and not conceptually justified. The basic computation to obtain a selection coefficient is to regress $\text{Log}(p/q)$ on time in a competition assay. In this regression, the slope corresponds to a selection coefficient (with the time unit that depends on the scaling chosen). The method chosen makes this computation somewhat contrived and dependent on absolute change in population size. Let us note (for simplicity) that final frequency p_f depends on selection coefficient s in a deterministic way. Taking weak selection for the sake of the argument, we have the standard pop gen equation $p_f = p_0 + s p_0(1-p_0)t$, where p_0 is the initial frequency. It is easy to show that $\text{Log}[p_f/(1-p_f)] - \text{Log}[p_0/(1-p_0)]$ is equal to s (at first order). This is the justification of the regression method just mentioned. However, the authors use a different approach. They first explain that they compute $m's = \text{Log}_2(N_f/N_0) / t$, then they compute s as $(m_i - m_j)/m_j$. It is straightforward to show that the latter quantity is not equal to s at first order, but to s divided by $\text{Log}(N_f/N_0)$. This might be non-consequential if this scaling is comparable in all competition experiments, but (1) this might cause some noise depending on the counts, (2) this is not conceptually justified (what is computed is not a selection coefficient), (3) this is unnecessary complicated. It is true that this problem has been carried over in many publications (mostly in experimental evolution), but there is no reason to use this approach and perpetuate this confusion.

Reviewer #3 (Remarks to the Author):

In this revised manuscript, Nieuwenhuis et al. significantly improve the clarity of the work and performed additional validation experiments to address my previous concerns. They have addressed all my major issues and I can recommend the manuscript for publication. Nevertheless, I list here a few minor suggestions that can further improve the clarity of the work. I don't think any interpretations here change their conclusions so I only list them as suggestions.

Clarifications suggested:

1) Line 179-198. I think get what the authors are trying to say, but the authors are being very confusing with terminology and scales

- "1.25 million zygotes per generation". They then multiply by 2.5 million zygotes in their

estimate, can they explain this logic in the text?

- "We can make a rough estimate of the upper limit of mutation rate [...] = $7.2 * 10^9$ mutation per sexual generations [...], but is much lower than [...] our maximum estimate of 10^6 per meiosis." These scales of comparisons are really confusing. The authors probably mean the 'lower limit'.

- Their test for a maximum estimate yielded zero switchers out of 10^6 meioses and the authors interpret that the upper bound must be $1/10^6$. I don't think that this is correct. The upper bound is better interpreted as being $\sim 3/10^6$ as the authors cannot reject that hypothesis based on their results.

- The lower limit of 1 per $7.2 * 10^9$ is calculated, but this limit is extremely conservative. The authors still ignore the selection effect of being a switcher during the asexual growth phase (not just drift). Given that, and the positive frequency dependence observed, I still think the authors need to tone down their conclusions about this event 'being rare' when this limit is so poorly defined, I could easily see this rate being $\sim 10^{-6}$ and that seems to be the scale that the users use for 'occurring readily'.

2) Figure 3, I suspect they are trying to emphasize the frequency dependence aspect. It might be worth trying to see if a figure with frequency on the x-axis to be better at representing this data, either as 4 panels (1 per dilution), or as one panel with 4 lines (1 per dilution).

3) Figure 2, there is a typo in the figure (inversion).

4) First mention of fission yeast, it might be good to mention the full organism name.

5) Line 50, mention self-compatible hermaphroditism.

6) Line 66 can be rephrased for clarity.

7) Line 57/58, 'in most fungi', and 'in most species', seem contradictory unless 'in most species' does not include fungi. This can be easily confused.

8) Line 105, mention lab strains 'of fission yeast'.

9) Figure 1, what are the branch lengths?

Reviewer #1 (Remarks to the Author):

My comments have been satisfactorily addressed by the authors; I don't have any remaining criticism and recommend acceptance.

Thank you for reviewing our manuscript anew and for your positive reaction.

Reviewer #2 (Remarks to the Author):

The revision is very well done. The text is clearer and to the point. The only place where the revision is still a bit unclear is related to the computation of selection coefficients. Otherwise, I'm happy with the revision. Congratulations for this!

Thank you for your very positive evaluation of the revised version of our manuscript.

The equations used are somewhat complicated and not conceptually justified.

The basic computation to obtain a selection coefficient is to regress $\text{Log}(p/q)$ on time in a competition assay. In this regression, the slope corresponds to a selection coefficient (with the time unit that depends on the scaling chosen). The method chosen makes this computation somewhat contrived and dependent on absolute change in population size. Let us note (for simplicity) that final frequency p_f depends on selection coefficient s in a deterministic way. Taking weak selection for the sake of the argument, we have the standard pop gen equation $p_f = p_0 + s p_0(1-p_0)t$, where p_0 is the initial frequency. It is easy to show that $\text{Log}[p_f/(1-p_f)] - \text{Log}[p_0/(1-p_0)]$ is equal to s (first order). This is the justification of the regression method just mentioned. However, the authors use a different approach. They first explain that they compute $m's = \text{Log}_2(N_f/N_0) / t$, then they compute s as $(m_i - m_j)/m_j$. It is straightforward to show that the latter quantity is not equal to s at first order, but to s divided by $\text{Log}(N_f/N_0)$. This might be non-consequential if this scaling is comparable in all competition experiments, but (1) this might cause some noise depending on the counts, (2) this is not conceptually justified (what is computed is not a selection coefficient), (3) this is unnecessary complicated. It is true that this problem has been carried over in many publications (mostly in experimental evolution), but there is no reason to use this approach and perpetuate this confusion.

We thank the reviewer for this clarification. We recalculated the selection coefficient as suggested by the reviewer for the asexual competitions. We calculated the selection coefficients as per generation selection coefficient.

For the sexual populations, the selection coefficient is too large for a first order approximation, hence we first calculate fitness using $\ln(w) = \ln(p_f / q_f) - \ln(p_0 / q_0)$ and then calculate the selection coefficient using: $w = 1 - s$. This results in selection coefficients against non-switchers, ranging between 0.30 and 0.84.

The figures 3, S8-S10 are modified accordingly, and the new calculations are explained in the methods section (lines 537-543 and 580-582). The new values do not affect the conclusion of our experiments.

Reviewer #3 (Remarks to the Author):

In this revised manuscript, Nieuwenhuis et al. significantly improve the clarity of the work and performed additional validation experiments to address my previous concerns. They have addressed all my major issues and I can recommend the manuscript for publication. Nevertheless, I list here a few minor suggestions that can further improve the clarity of the work. I don't think any interpretations here change their conclusions so I only list them as suggestions. Clarifications suggested:

1) Line 179-198. I think get what the authors are trying to say, but the authors are being very confusing with terminology and scales
- "1.25 million zygotes per generation". They then multiply by 2.5 million zygotes in their estimate, can they explain this logic in the text?

We addressed this valuable comment and carefully clarified this in the text. The logic is that in each zygote there are two genomes that each can obtain the alternative mating type genes from the other genome. The total number of zygotes is thus 1.25 million and the number of chromosomes is two times that.

- "We can make a rough estimate of the upper limit of mutation rate [...] = $7.2 * 10^9$ mutation per sexual generations [...], but is much lower than [...] our maximum estimate of 10^6 per meiosis." These scales of comparisons are really confusing. The authors probably mean the 'lower limit'.

We changed the text such that our empirical observations are consistently referred to as the upper limit of the mutation rate and our calculations are the lower limit or the mutation rate.

- Their test for a maximum estimate yielded zero switchers out of 10^6 meioses and the authors interpret that the upper bound must be $1/10^6$. I don't think that this is correct. The upper bound is better interpreted as being $\sim 3/10^6$ as the authors cannot reject that hypothesis based on their results.

This is a good point. In our previous calculations we did not correct for a sampling bias. We added the calculation to the text "This corresponds to a maximum mutation rate μ of $3 * 10^{-6}$ under binomial sampling ($P(K=0; N=106; p=3 * 10^{-6}) < 0.05$) which is clearly not sufficient to explain recurrent fixation in 9 population." (lines 182-184)

- The lower limit of 1 per $7.2 * 10^9$ is calculated, but this limit is extremely conservative. The authors still ignore the selection effect of being a switcher during the asexual growth phase (not

just drift). Given that, and the positive frequency dependence observed, I still think the authors need to tone down their conclusions about this event ‘being rare’ when this limit is so poorly defined, I could easily see this rate being $\sim 10^{-6}$ and that seems to be the scale that the users use for ‘occurring readily’.

We agree that this is a conservative estimate and hence we reduced the emphasis on a low mutation rate. Additionally, we now explicitly explain that during the asexual phase the sampling of 1% of the cells after growth will reduce the chance of sampling this strain. “Note that this is a conservative estimate, as it ignores drift during the asexual growth phase. A novel switching genotype might be lost when one percent of the population is subsampled after growth with a chance of $P(K=0; N=5 \cdot 10^8; p=5 \cdot 10^{-8}) = 0.37$ under binomial sampling, resulting in a final estimate of $1.14 \cdot 10^{-8}$ mutations per sexual generation. This estimate lies far above the per generation mutation rate for mitotic division of 10^{-10} (ref. 47) but is much lower than the spontaneous rate of reversion from the non-switching h^{+N} genotype to switching of $9 \cdot 10^{-5}$ per mitosis (ref. 29) and our upper limit of $3 \cdot 10^{-6}$ per meiosis.” (lines 197-203).

Because the cost of switching has not yet been introduced in the text when we discuss the mutation rate, we omitted the cost of switching in the calculations. However, this does not greatly affect the lower limit of the mutation rate. Binomial sampling assuming a cost of 3% for 6.6 generations results in a chance of 0.44 for the mutation to be lost and even with a very strong selection coefficient of 10% this chance is still only 0.53, resulting in a mutation rate around $1.25 \cdot 10^{-8}$ to $1.5 \cdot 10^{-8}$.

The frequency dependence as mentioned by the reviewer probably does not apply to the density used in the evolution experiment. As can be seen in figure 3a and better in the new supplementary figure S9 (see also next comment), the frequency dependence is lost and possibly reversed at low densities.

2) Figure 3, I suspect they are trying to emphasize the frequency dependence aspect. It might be worth trying to see if a figure with frequency on the x-axis to be better at representing this data, either as 4 panels (1 per dilution), or as one panel with 4 lines (1 per dilution).

The main message of this figure is to show that density is the main factor driving the benefit of selection, whereas frequency dependence has less of an effect. Within each panel, the effect of density is clearly visible, and it is only slightly affected by the initial frequencies. We therefore prefer to keep the figure as it is. We did add a figure as suggested as supplementary figure 9.

3) Figure 2, there is a typo in the figure (inversion).

Thank you for spotting this typo, which we corrected.

- 4) First mention of fission yeast, it might be good to mention the full organism name.

We now mention the full name.

- 5) Line 50, mention self-compatible hermaphroditism.

We added 'self-compatibility' to this sentence.

- 6) Line 66 can be rephrased for clarity.

We rephrased the sentence to "Note that reproductive assurance is not necessarily the only possible explanation for the evolution of homothallism¹⁸. Alternative reasons may be a reduction in outbreeding depression¹⁹ or an increase in compatibility to facilitate outcrossing², and these three explanations are not mutually exclusive^{13,20}." (Lines 64-68)

- 7) Line 57/58, 'in most fungi', and 'in most species', seem contradictory unless 'in most species' does not include fungi. This can be easily confused.

The sentence has been modified to: "Even though most fungi are able to reproduce asexually, sexual reproduction is often an obligatory phase of the lifecycle requiring conjugation¹²" (lines 56-57)

- 8) Line 105, mention lab strains 'of fission yeast'.

Changed as suggested.

- 9) Figure 1, what are the branch lengths?

The branch length and scale bar is in "substitutions per site". Information was added to figure captions.

REVIEWERS' COMMENTS:

Reviewer #2 (Remarks to the Author):

I'm happy with this second revision. No additional comment.

Reviewer #3 (Remarks to the Author):

The authors have adequately revised their manuscript and I recommend the article for publishing.